# Reg4 and complement factor D prevent the overgrowth of *E. coli* in the mouse gut

Houbao Qi[1,2,3], Jianmei Wei[1,2,3], Yunhuan Gao[1,2,3], Yazheng Yang [1,2,3], Yuanyuan Li[1,2,3], Hua Zhu[4], Lei Su[4], Xiaomin Su[1,2,3], Yuan Zhang[1,2,3] & Rongcun Yang [1,2,3 ✉]

The expansion of *Enterobacteriaceae*, such as *E. coli* is a main characteristic of gut inflammation and is related to multiple human diseases. However, how to control these *E. coli* overgrowth is not well understood. Here, we demonstrate that gut complement factor D (CFD) plays an important role in eliminating *E. coli*. Increased *E. coli*, which could stimulate inflammatory macrophages to induce colitis, were found in the gut of CFD deficient mice. We also showed that gut Reg4, which is expressed in gut epithelial cells, stimulated complement-mediated attack complexes to eliminate *E. coli*. *Reg4* deficient mice also had increased *E. coli*. The dominant *E. coli* were isolated from colitis tissues of mice and found to be sensitive to both CFD- and Reg4-mediated attack complexes. Thus, gut Reg4- and CFD-mediated membrane attack complexes may maintain gut homeostasis by killing inflammatory *E. coli*.

---

[1] State Key Laboratory of Medicinal Chemical Biology, Nankai University, Tianjin 300071, China. [2] Key Laboratory of Bioactive Materials Ministry of Education, Nankai University, Tianjin 300071, China. [3] Department of Immunology, Nankai University School of Medicine; Nankai University, Tianjin 300071, China. [4] Key Laboratory of Human Disease Comparative Medicine, Ministry of Health, Institute of Laboratory Animal Science, Chinese Academy of Medical Sciences (CAMS) Comparative Medical Center, Peking Union Medical College (PUMC), 100021 Beijing, China. ✉email: ryang@nankai.edu.cn

The expansion of facultative anaerobic *Enterobacteriaceae* (phylum *Proteobacteria*) with a high level of commensal *Escherichia coli* (*E. coli*) is a common characteristic during inflammation in the colon[1–4]. These increased *E. coli* levels are related to the occurrence and development of multiple diseases such as obesity, diabetes, cardiaovascular and liver diseases, and tumor[5–9]. Interactions between the microbiota and pathogenic bacteria[10], microcin-mediated competition among *Enterobacteriaceae* in the inflamed gut[11], microbiota-activated PPAR-γ signaling[12], dietary zinc[13] and tungstate-mediated microbiota editing[14] may affect the expansion of these bacteria. However, it is incompletely clear how to control these increased *E. coli*.

The membrane attack complex (MAC), a hetero-oligomeric protein assembly that kills pathogens by perforating their cellular envelopes, has an essential role in human immunoprotection against gram-negative bacteria such as *E. coli*[15–17]. The formation of the MAC depends on the sequential assembly of the soluble complement proteins C5b, C6, C7, C8, and C9, and is mainly triggered by immune complexes or pattern recognition molecules (PRMs) upon recognition of non-self or altered self-cells, such as by C1q, collectins, ficolins, and properdin[18]. The MACs are also activated by many other factors such as pentraxins, c-reactive protein, serum-amyloid P component, and pentraxin 3[19]. Complement factor D (CFD), a specific serine protease that cleaves its unique substrate to generate the C3 convertases C3(H2O)Bb and C3bBb, may amplify all complement-mediated bactericidal effects, and play a critical role in mannose-binding lectin-mediated MACs[20–22]. However, many pathogenic organisms also find ways to escape complement attach through a range of different mechanisms such as Pic, extracellular serine protease P, the Adr2 protein and the C3b-fH complex[18,23]. Whether increased *E. coli* in colitis can be eliminated by MACs, and how to stimulate the formation of MACs in gut tissues remain elusive.

To prevent overgrowth of the commensal population and prohibit pathogens from colonizing and causing infection, the host must maintain tight control over gut homeostasis. Gut epithelial cells produce and release a variety of biomolecules such as defensins, lectins, mucins, and secretory immunoglobulin A into the mucosa and lumen to contribute to immunity[24]. In this study, we found that Reg4 (REG4 in humans), a member of the C-lectin family, which is expressed by gut Paneth cells or deep crypt secretory cells[25], may stimulate the complement system to kill inflammatory *E. coli*. We also demonstrated that gut epithelial-derived CFD plays a critical role in eliminating gut *E. coli*. Thus, both gut Reg4 and CFD may induce bactericidal activity against *E. coli* to maintain gut homeostasis.

## Results

### CFD^fl/fl^pvillin-cre^T^ mice are sensitive to DSS-mediated colitis through the accumulation of inflammatory macrophages in gut tissues.

CFD was detected in Paneth cells in the small intestine and Paneth cell-like cells in the colon of mice (Fig. 1a, b), consistent with our and other data[26,27]. We also found that the gut microbiota may regulate the expression of CFD in gut epithelial cells. The levels of CFD in the small intestine and colon in germ-free (GF) mice were markedly lower than those in wild-type (*wt*) mice (Supplementary Fig. 1a–d). The colonization of mouse feces, *E. coli* or *lactobacilli* in GF mice could promote the expression of CFD (Supplementary Fig. 1e, f). Since CFD may be involved in killing gram-negative bacteria such as *E. coli*, which are related to colitis[15–17], CFD expression in gut epithelial cells may play a role in the occurrence and development of colitis. To investigate this, we generated gut *CFD* conditional knockout mice (CFD^fl/fl^pvillin-cre^T^mice) along with *wt* littermate control CFD^fl/fl^pvillin-cre^w^mice according to the described strategy. In CFD^fl/fl^pvillin-cre^T^mice, complement factor D

(CFD, Gene ID: 5068) was selectively knocked out in mouse gut epithelial cells (Fig. 1a, b). We next used dextran sulfate sodium (DSS)-mediated model to determine sensitivity of CFD^fl/fl^pvillin-cre^T^mice to colitis. We found that CFD^fl/fl^pvillin-cre^T^mice were more sensitive to 2.5% DSS induced colitis than their control littermate wild-type CFD^fl/fl^pvillin-cre^w^mice, and exhibited lower survival rate, more weight loss, higher disease indexes, more colonic shortening and higher histological score (Fig. 1c–g). Higher levels of the inflammatory cytokines TNFα, IL6 and ILβ were also detected in the colonic tissues of CFD^fl/fl^pvillin-cre^T^mice. Gut tissue inflammatory macrophages play a critical role in occurrence and development of colitis[28–31]. These cells may be characterized as CD45+ CD11b+CD103− CD64+ MHCII+ly6C+ cells[17,18]. Increased inflammatory macrophages were observed in both CFD^fl/fl^pvillin-cre^T^mice and especially DSS-treated CFD^fl/fl^pvillin-cre^T^mice (Fig. 1i–j), suggesting that the sensitivity of CFD^fl/fl^pvillin-cre^T^mice to DSS-mediated colitis may depend on inflammatory macrophages. Taken together, CFD^fl/fl^pvillin-cre^T^mice are sensitive to DSS-mediated colitis through the markedly accumulation of inflammatory macrophages in gut tissues.

### *E. coli* levels are markedly increased in CFD^fl/fl^pvillin-cre^T^ mice.

Since CFD may promote bactericidal activity (Fig. 2a), we next analyzed the composition of the gut microbiota. Sequencing analyses of 16S ribosomal RNA (V3-V4 variable region) in colonic contents did not show marked changes in the composition such as the proportion of *Proteobacteria* under normal physiological conditions in CFD^fl/fl^pvillin-cre^T^mice and cohoused control CFD^fl/fl^pvillin-cre^w^mice (Fig. 2b, and Supplementary Fig. 3a). However, after DSS treatment, which may cause relative increase in the luminal abundance of *Enterobacteriaceae* (phylum *Proteobacteria*) (Supplementary Fig. 3b)[5], there were markedly increased levels of gut *Enterobacteriaceae*, mainly *E. coli* in the colonic contents of CFD^fl/fl^pvillin-cre^T^mice as compared to those of CFD^fl/fl^pvillin-cre^w^ mice (Fig. 2c, d, and Supplementary Fig. 3c), suggesting that CFD deficiency in gut tissues causes reduced killing on *E. coli* (*Escherichia-Shigella*). The increased *E. coli* were confirmed by flow cytometry, qRT-PCR and fluorescence hybridization (FISH) (Fig. 2e–g). q-PCR also showed that only the proportion of *E. coli* but not level of total bacteria changed in DSS-treated CFD^fl/fl^pvillin-cre^T^mice (Fig. 2f). Thus, CFD deficiency not only changes the composition of the gut microbiota but also promotes the abundance of *E. coli* in colitis.

### *E. coli* promote sensitivity to DSS-mediated colitis and the accumulation of inflammatory macrophages in CFD^fl/fl^pvillin-cre^T^ mice.

We next determined whether the increased *E. coli* promoted the sensitivity of CFD^fl/fl^pvillin-cre^T^mice to DSS-mediated colitis. We identified the dominant *E. coli* in DSS-treated mice, named *E. coli* 0160[32]. After equal amount of *E. coli* 0160 were infused into mice, CFD^fl/fl^pvillin-cre^T^mice had more severe colitis than CFD^fl/fl^pvillin-cre^W^mice after the administration of 2% DSS, including lower survival rate, more weight loss, higher disease indexes and more colonic shortening (Fig. 3a–d). Histological H/E staining and the expression of inflammatory cytokines also verified the worsened inflammation in CFD^fl/fl^pvillin-cre^T^mice after the infusion of *E. coli* (Fig. 3e, f). Since *E. coli* potentially promoted colitis, we selected a low concentration (2% DSS). Taken together, CFD^fl/fl^pvillin-cre^T^mice are more sensitive to DSS-mediated colitis after the infusion of *E. coli*. Notably, more accumulated inflammatory macrophages were observed in the colon tissues of CFD^fl/fl^pvillin-cre^T^ mice (Fig. 3g, h) after the infusion of *E. coli*. Gram-negative pathogenic bacteria may directly activate not only inflammatory macrophages but also gut epithelial cells to produce IL-18 and induce the

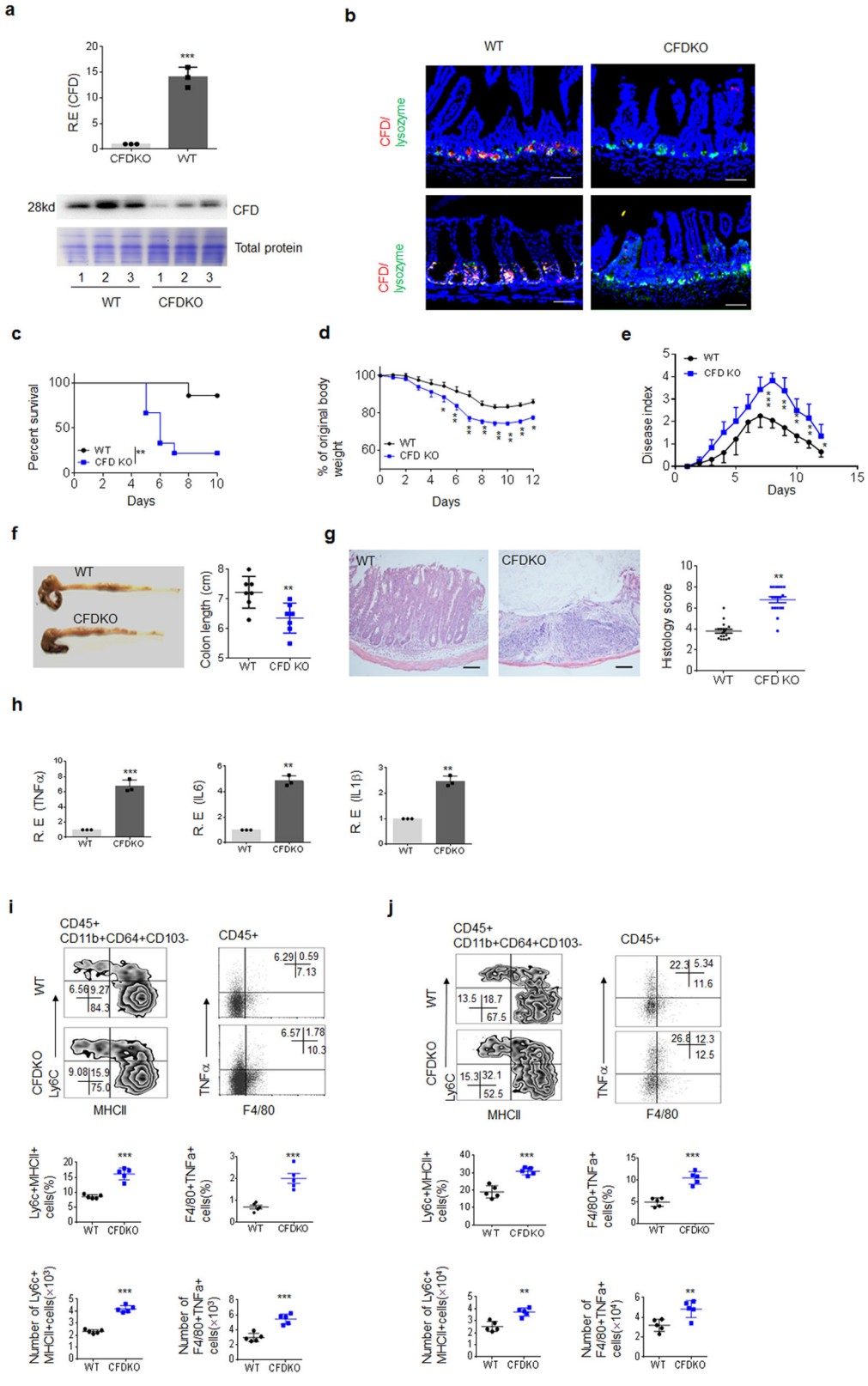

macrophages through IFNγ producing cells[33]. ELISA and immunostaining showed markedly increased IL-18 in gut epithelial cells in DSS-treated CFD$^{fl/fl}$pvillin-cre$^T$ mice (Supplementary Fig. 4a, b). Increased levels of mature IL-18 were also present in the colonic tissues of CFD$^{fl/fl}$pvillin-cre$^T$ mice that were treated

ex vivo in *E. coli* stimulation (Supplementary Fig. 4c–e). Gram-bacteria such as *Salmonella* may activate macrophages and gut epithelial cells through NLRC4, caspase11 etc[34–37]. We found that NLRC4, caspase-1/11, caspase 8 and PKCδ in gut epithelial cells of CFD$^{fl/fl}$pvillin-cre$^T$ mice were involved in the expression of

**Fig. 1 CFD-deficient mice are more sensitive to DSS-mediated colitis and have marked accumulation of inflammatory macrophages in colonic tissues.**
**a** qRT-PCR (upper) of CFD in colon tissues and immunoblotting (lower) of CFD in equal amount of colon contents of CFD$^{fl/fl}$pvillin-cre$^W$(WT) and CFD$^{fl/fl}$pvillin-cre$^T$mice (CFDKO). Total protein was stained using Coomassie. Number, different individuals. **b** Immunostaining of CFD in ileum (upper) and colon (lower) of CFD$^{fl/fl}$pvillin-cre$^W$mice (WT) and CFD$^{fl/fl}$pvillin-cre$^T$mice (CFDKO). Red, CFD; Green, lysozyme. Scale bar, 40 μM. **c, d** and **e** The survival rate (C), body weight (D) and disease index (E) in CFD$^{fl/fl}$pvillin-cre$^T$mice (CFDKO) and CFD$^{fl/fl}$pvillin-cre$^W$ (WT) mice (male, $n = 15$) after DSS (2.5%) treatment. **f** The length of colon in CFD$^{fl/fl}$pvillin-cre$^T$mice (CFD KO) and CFD$^{fl/fl}$pvillin-cre$^W$mice (WT) after DSS (2.5%) treatment ($n = 6$). **g** H&E staining and histology score of colon tissues of CFD$^{fl/fl}$pvillin-cre$^T$mice (CFDKO) and CFD$^{fl/fl}$pvillin-cre$^W$mice (WT) after DSS (2.5%) treatment. For histological score, three slides/mouse, $n = 6$. Scale bar, 40 μM. **h** QRT-PCR of inflammatory cytokines TNFα, IL-6 and IL-1β in the colonic tissues of CFD$^{fl/fl}$pvillin-cre$^T$ (CFDKO) and CFD$^{fl/fl}$pvillin-cre$^W$mice (WT) after DSS (2.5%) treatment ($n = 6$). **i** Flow cytometry of CD45$^+$CD11b$^+$CD64$^+$CD103$^-$MHC$^+$Ly6C$^+$, F4/80$^+$TNFα$^+$ cells in colon tissues in normal CFD$^{fl/fl}$pvillin-cre$^T$ (CFD KO) and CFD$^{fl/fl}$pvillin-cre$^w$ (WT) mice ($n = 5$). **j** Flow cytometry of CD45$^+$CD11b$^+$CD64$^+$CD103$^-$MHC$^+$Ly6C$^+$, F4/80$^+$TNFα$^+$ cells in the colon tissues of DSS-treated CFD$^{fl/fl}$pvillin-cre$^T$ (CFDKO) and CFD$^{fl/fl}$pvillin-cre$^w$ (WT) mice. Kruskal–Wallis test in **c**; Analysis of variance test in **d** and **e**; Two side student's $t$ test in **f** and **h–j**; Mann–Whitney $U$ test in **g**; *$P < 0.05$; **$P < 0.01$; ***$P < 0.001$; NS no significance. Data in **c–e** are a representative of three independent experiments. R.E relative expression.

IL-18, which was induced by the administration of *E. coli* 0160 (Supplementary Fig. 4c–e). Since the generation of inflammatory macrophages in gut tissues is dependent on IFNγ[33], we also analyzed the IFNγ-producing cells. A drastic expansion of interferon γ (IFNγ)-producing CD4$^+$ T helper cells (CD4$^+$IFNγ$^+$ Th1 cells) was observed in *E. coli* infused CFD$^{fl/fl}$pvillin-cre$^T$mice (Fig. 3h). NK cells, another IFNγ producing cell type, also accumulated in gut tissues of *E. coli* infused CFD$^{fl/fl}$pvillin-cre$^T$mice (Fig. 3h). Thus, our data indicate that *E. coli* promote not only colitis but also the accumulation of inflammatory macrophages in the gut tissues of CFD$^{fl/fl}$pvillin-cre$^T$mice.

**CFD$^{fl/fl}$pvillin-cre$^T$ mice have reduced bactericidal abilities against *E. coli*.** We next examined whether CFD$^{fl/fl}$pvillin-cre$^T$mice had reduced bactericidal abilities against *E. coli*. We first generated GFP-labeled *E. coli* 0160. When equal amounts of GFP-labeled *E. coli* were infused into CFD$^{fl/fl}$pvillin-cre$^T$ and CFD$^{fl/fl}$pvillin-cre$^W$ mice, the fluorescence intensities in the small intestine, colon and cecum were higher in CFD$^{fl/fl}$pvillin-cre$^T$mice than in CFD$^{fl/fl}$pvillin-cre$^W$ mice (Fig. 4a). The number of *E. coli* in the colon tissues and feces of CFD$^{fl/fl}$pvillin-cre$^T$mice was much higher than that in CFD$^{fl/fl}$pvillin-cre$^W$ mice (Fig. 4b). These results suggest that CFD$^{fl/fl}$pvillin-cre$^T$mice have reduced bactericidal abilities against *E. coli*. To further determine whether the increase in *E. coli* was from CFD deficiency in CFD$^{fl/fl}$pvillin-cre$^T$ mice, we generated CFD$^{fl/fl}$pvillin-cre$^T$ and CFD$^{fl/fl}$pvillin-cre$^W$ germ-free (GF) mice. When equal amounts of GFP-labeled *E. coli* were infused into CFD$^{fl/fl}$pvillin-cre$^T$ and CFD$^{fl/fl}$pvillin-cre$^W$ GF mice, the fluorescence intensity and number of *E. coli* in CFD$^{fl/fl}$pvillin-cre$^T$ GF mice were also higher than those in CFD$^{fl/fl}$pvillin-cre$^W$ GF mice (Fig. 4c, d). Flow cytometry also showed fewer dead bacteria in CFD-KO mice (Fig. 4e). Since we mainly examined the killing effect of CFD on *E. coli* in the gut cavity, these mice were not treated with DSS.

Notably, when equal amounts of GFP-labeled E. coli were infused into mice, there was reduced C3b deposition on *E. coli* in CFD$^{fl/fl}$pvillin-cre$^T$ mice as compared to CFD$^{fl/fl}$pvillin-cre$^W$ mice (Fig. 4f). Metabolic factors were also different in colonic tissues from CFD$^{fl/fl}$pvillin-cre$^T$ and CFD$^{fl/fl}$pvillin-cre$^W$ mice such as that C1, which is involved in the typical pathway; complement factor B (BF), which serves as the catalytic subunit of complement C3 convertase in the alternative pathway (AP); and C3 or the C5bC9 complex, which participate in all three pathway (Fig. 4g). More FB, and less C3b and C5bC9 were found in colonic tissues of CFD$^{fl/fl}$pvillin-cre$^T$ mice than in CFD$^{fl/fl}$pvillin-cre$^W$ mice; whereas C1q was not significantly changed in any analyzed samples from both types of mice (Fig. 4h). Reduced 5b-9 and C3b, and increased FB were also measured in the feces of CFD$^{fl/fl}$pvillin-cre$^T$ mice (Fig. 4i). Thus, these findings suggest that

there is decreased atypical complement-mediated bactericidal activity in CFD$^{fl/fl}$pvillin-cre$^T$ mice.

**Gut Reg4 initiates the membrane attack complex against *E. coli*.** We next investigated how the complement system killed *E. coli* in gut tissues. To do this, we first measured the sensitivity of *E. coli* to normal serum, C3-, C1- and CFD-deficient sera. We found that C3 deficiency markedly reduced *E. coli* killing as compared to that of C1 deficient serum (Fig. 5a). CFD-deficient serum also exhibited reduced killing of *E. coli* (Fig. 5b). Thus, *E. coli* isolated from colitis tissues may be sensitive to an unknown C3-dependent complement-mediated killing pathway. Lectin may induce the C3-dependent complement pathway through binding with the mannose in LPS such as the "O" antigen of mannose[38]. Since *E. coli* 0160 is "O" antigen positive[32], we performed lipopolysaccharide (LPS) and mannan blocking experiments. We found that LPS from *E. coli* (0111B4) containing "mannose"[39] and mannan but not peptidoglycan could inhibit normal serum-mediated killing on *E. coli* 0160 (Fig. 5c). We next examined which kind of lectin was involved in the complement-mediating killing of *E. coli*. There are several mannose-binding lectins such as mannose-binding lectin (MBL), ficolins (ficolin-1 or M-ficolin, ficolin-2 or L-ficolin, and ficolin-3 or H-ficolin), and collectin-11, which potentially activate complement pathways[40,41], but Reg 4 has a typical lectin fold, which is produced by gut epithelial cells, and may potentially bind with mannose in LPS in gut tissues[42]. We generated recombinant mouse Reg4 and human REG4 proteins, and examined their effects on *E. coli* killing (Supplementary Fig. 5). Both Reg4 and REG4 may promote killing by normal serum but not CFD-deficient sera; Furthermore this killing was dose-dependent (Fig. 5d–f). However, mutant hREGIV-P91S, which has comparably reduced mannan-binding abilities[42], only slightly promoted to complement-mediated *E. coli* killing, suggesting that Reg4 in gut tissues participates in *E. coli* killing (Fig. 5d). IgA, a major of defense factor, was also involved in the lectin-mediated activation of complement, consistent with other studies[43–45] (Fig. 5g). To further determine the effects of Reg4 and IgA, we measured the deposition of these components on *E. coli* in vivo. When equal amounts of GFP-labeled *E.coli* were infused into 2% DSS-treated CFD$^{fl/fl}$pvillin-cre$^T$ or CFD$^{fl/fl}$pvillin-cre$^w$ mice, which were treated with pan-antibiotics to eliminate the effects of other bacteria on *E. coli*, we found that Reg4 and IgA bound to *E. coli* in colon tissues (Fig. 6a). These *E. coli* also bound to MBL-associated serine protease (MASP)1/3 (Fig. 6a), which triggers complement activation through serine proteases[46,47]. However, *E. coli* did not bind with WASP2[48]. Flow cytometry also showed the binding of *E. coli* with Reg4 and IgA, but not collectin, which is also involved in mediating complement activation[41] (Fig. 6b).

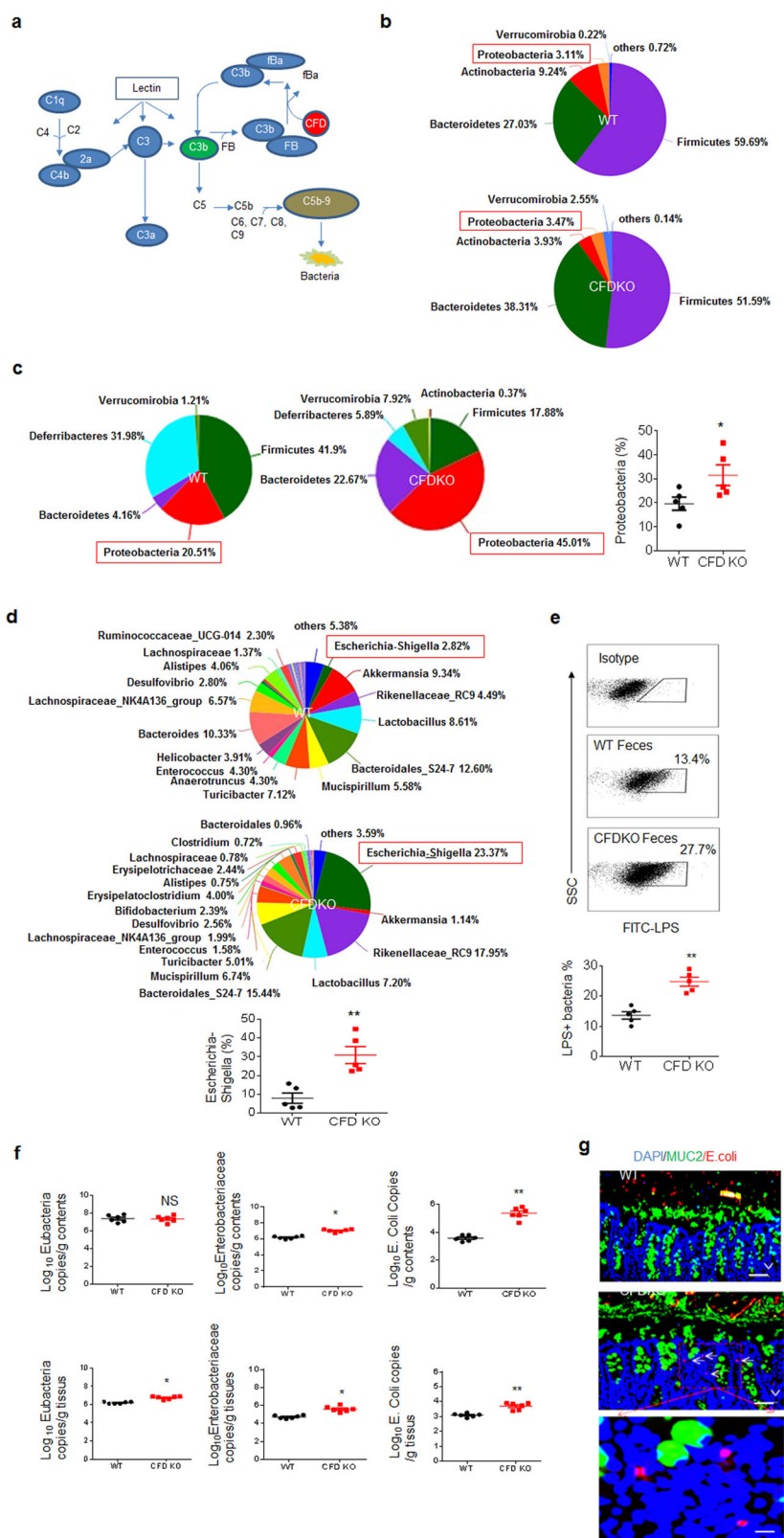

Finally, we used ELISA to examine the binding of Reg4 with LPS, we found that Reg4 but not control BSA (bovine serum albumin) could bind with LPS; Furthermore this binding was dose-dependent (Fig. 6c). We also confirmed the binding of LPS with IgA[44] (Fig. 6c). Reg4 or IgA caused the deposition of C3 and the combination of two factors caused more deposition of C3 than either factor alone (Fig. 6d, e). These data indicate that Reg4 initiate MAC activation to kill *E. coli*. Taken together, these data suggest that Reg4 may be involved in the initiation of complement-mediated MACs killing of *E. coli*.

**Fig. 2 E. coli markedly increase in CFD$^{fl/fl}$pvillin-cre$^T$ mice. a** Map of complement-mediated bactericidal activity. **b** The proportion of gut bacteria after 16s rRNA analyses of colon contents in normal CFD$^{fl/fl}$pvillin-cre$^W$ (WT) and CFD$^{fl/fl}$pvillin-cre$^T$ (CFDKO) mice (pooled samples from six mice). Also see Fig. S3A. **c** The proportion of gut bacteria after 16s rRNA analyses in CFD$^{fl/fl}$pvillin-cre$^W$ (WT) and CFD$^{fl/fl}$pvillin-cre$^T$ (CFDKO) mice after 2% DSS. Also see Fig. S3B. **d** The proportion of E. coli (Escherichia-Shigella) in CFD$^{fl/fl}$pvillin-cre$^W$ (WT) and CFD$^{fl/fl}$pvillin-cre$^T$ (CFDKO) mice after 2% DSS ($n = 5$). Also see Fig. S3C. **e** Flow cytometry of feces bacteria after staining using anti-LPS antibodies in CFD$^{fl/fl}$pvillin-cre$^W$ (WT) and CFD$^{fl/fl}$pvillin-cre$^T$ (CDKO) mice after 2% DSS. **f** QPCR of colonic content (upper) and colonic tissues (lower) in CFD$^{fl/fl}$pvillin-cre$^W$ (WT) and CFD$^{fl/fl}$pvillin-cre$^T$ (CFDKO) mice after 2% DSS ($n = 6$). **g** FISH of E. coli in the gut tissues of CFD$^{fl/fl}$pvillin-cre$^W$ (WT) and CFD$^{fl/fl}$pvillin-cre$^T$ (CFDKO) mice after 2% DSS. One representative from six mice. Scale bar, 40 μM; Scale bar in bottom panel, 5 μM. Two side student's $t$ test in **e** and **f**; *$P < 0.05$; **$P < 0.01$; ***$P < 0.001$; NS no significance.

**Reg4 deficient mice have reduced complement bactericidal abilities against E. coli.** To further determine the role of gut Reg4 in complement-mediated killing on E. coli, we generated Reg4 deficient mice according to described strategy (Supplementary Fig. 6a–c). Since E. coli may accumulate in DSS-mediated colitis[5], we analyzed and compared the proportion of E. coli in Reg4 deficient and their cohoused control wt mice after 2% DSS-treatment. 16S rRNA analyses demonstrated that there were increased proteobacteria in Reg4 deficient mice as compared to the cohoused control wt mice (Fig. 7a, b, and Supplementary Fig. 6d). Equal amount of GFP-labeled E. coli 0160 were administered to pan-antibiotics-treated mice, and the fluorescence intensities in the small intestine, colon and cecum were higher in Reg4 deficient mice than control mice, suggesting reduced bactericidal abilities in these mice (Fig. 7c). There was increased E. coli in feces collected at different times (Fig. 7d). Flow cytometry also showed fewer dead bacteria in Reg4 deficient mice than control mice (Fig. 7e). Flow cytometry also exhibited less deposition of C3b on the bacteria in Reg4 deficient mice than control mice (Fig. 7f). The data also further confirmed tht binding of IgA and Reg4 but not collectin to GFP-labeled E. coli (Fig. 7f). Reduced C3b and C5bc9, and increased FB in the colon tissues and feces of Reg4 deficient mice was also observed as compared to those of the control wt mice (Supplementary Fig. 7a, b). Finally, the activity of CFD in Reg4 deficient mice was not different from that in wt mice (Supplementary Fig. 8). Taken together, these data suggest that Reg4 deficient mice exhibit reduced complement-mediated killing of E. coli.

## Discussion

Inflammatory diseases of the gastrointestinal tract are frequently associated with changes in gut microbial communities that include the expansion of facultative anaerobic bacteria (Phylum Proteobacteria, family Enterobacteriaceae, mainly E. coli),which is a common characteristic of dysbiosis[2–4,8,49]. These bacteria are related to gut inflammation, and are more importantly associated with other human diseases such as obesity, tumor and liver diseases. Here, we showed that complement-mediated MAC plays an important role in eliminating gut E. coli. We found that gut-derived Reg4 and/or IgA may initiate the formation of MACs to kill E. coli. Thus, our data demonstrate that both gut-derived Reg 4 and CFD are involved in maintaining gut homeostasis. These findings pave the way for the development of therapeutic strategies for gut inflammation-associated diseases.

The complement system, a proteolytic cascade, can be activated via the classic pathway (CP), the lectin pathway (LP), and the alternative pathway (AP). Complement factor D (CFD) may amplify all complement-mediated bactericidal abilities, and play an important role in mannose-binding lectin-mediated MACs that target bacteria[20–22]. CFD may be derived from multiple kinds of cells, especially those in adipose tissues. Here, we demonstrated that gut epithelial cell-derived CFD plays a critical role in eliminating increased E. coli in colitis.

We found that Reg4 initiated complement-mediated killing of E. coli. The Lectin family is composed of carbohydrate-binding proteins that contribute to mucosal innate immunity. Several lectins such as mannose-binding lectin (MBL) and ficolin1, 2 and 3 are pattern recognition molecules that are involved in the initiation of complement activation[40]. These pathways are involved in the protecting against E. coli infection[50]. Other lectins such as REG3α in humans and its murine homolog REG3γ are also found throughout small intestinal epithelium[51]. The interactions between REG3α and bacterial targets may kill bacteria through peptidoglycan. Reg 4 may bind to polysaccharides, mannose, and heparin in the absence of calcium[42]. We demonstrate that Reg 4 may bind with E. coli LPS to initiate the formation of MACs. Additionally, Reg4 may also bind mannose at two calcium-independent sites and directly induce damage to the bacterial cell wall[42,52].

IgA may promote Reg4-mediated complement membrane attack complexes. Upon interaction of IgA with pathogens, IgA molecules can have diverse effector functions, including directly preventing the invasion of microorganisms, the interaction with the phagocytic IgA Fc receptor, and complement activation. Secretory IgA (sIgA) is the dominant humoral immunity at the intestinal mucosa. Due to its polymeric nature and multivalency, sIgA primarily protects mucosal surfaces by noncovalently crosslinking microorganisms or macromolecules. sIgA also plays a role in shaping and maintaining the gut microbiota from birth to adulthood[53]. Here we demonstrated that IgA and Reg4 may initiate the activation of complement pathways. Polymeric IgA has recently been suggested to activate the lectin pathway and contribute to complement activation[44]. Complement activation by IgA was previously shown to involve the alternative, but not the classical complement pathway[43,44].

Gut mucosal immunity includes a complex network of different types of cells that work together to provide physical and chemical barriers against microbial invasion, and maintain gut homeostasis. Paneth cells can produce a variety of peptides and proteins including Reg4, CFD, α-defensins, lysozyme C, phospholipases and the C-type lectin Reg3α[54]. The degranulation and expression of these proteins occurs in response to stimuli including cholinergic agonists, bacteria, and bacterial products such as lipopolysaccharide (LPS) and lipoteichoic acid[55]. Our results also demonstrate that the expression of CFD may be regulated by gut bacteria. These small molecules may also be synthesized in vitro. Since these components may be regulated and synthesized in vitro, our findings have implications for improving strategies to eliminate E. coli.

## Methods

All reagents and oligoes used in this study were listed in Supplementary Table 1.

**Mice**. Four-to six-week-old male or female C57BL/6 mice were obtained from Nanjing Animal Center. Caspase-1/Caspase 11−/−, and NLRC4−/− in B6 background from Prof. Meng in University of Chinese Academy of Sciences, shanghai and Prof. Shao in National Institute of Biological Sciences, Beijing were bred and kept under specific pathogen-free (SPF) condition in Nankai University.

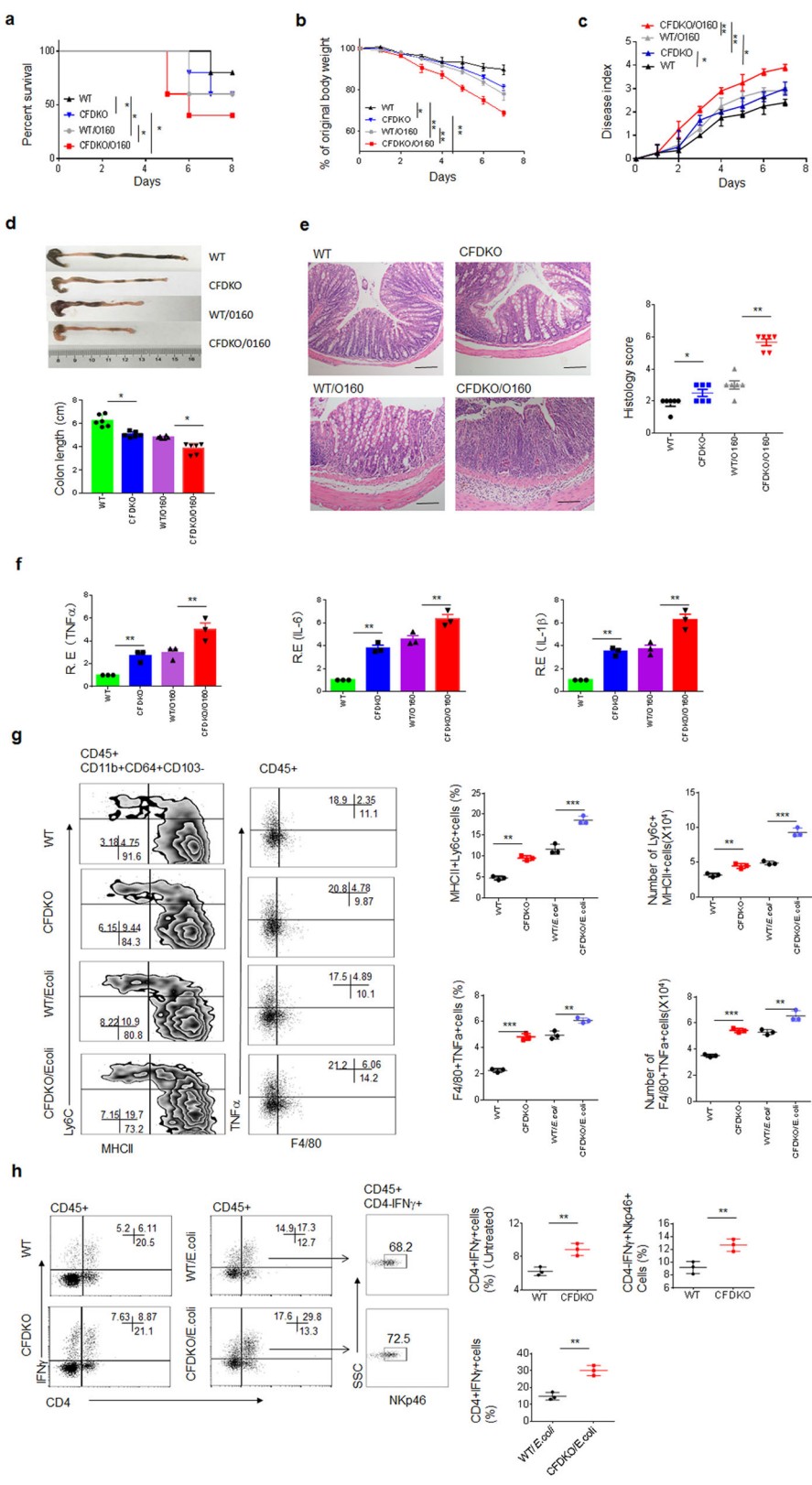

C57BL/6 germ-free (GF) mice were generated by Beijing Animal Center. All experiments in GF mice were performed in Institute of Laboratory Animal Science, Chinese Academy of Medical Sciences (CAMS) Comparative Medical Center, Peking Union Medical College (PUMC), Beijing 100021, China.

CFD$^{fl/fl}$pvillin-cre$^T$mice (gut CFD conditional knockout mice) were prepared by NBRI, China according to the described strategy in Supplementary Fig. 2. Transcript CFD201(ENSMUST00000061653) is 0.923 Kb long and composed

of 5 exons encoding a 259 aa protein. The first loxP site was put in downstream of Exon1, and the second loxP site in downstream of the Exon5; Exon2-5 (≈1201 bp) were floxed with loxP site, and removed via Cre/LoxP excision. All domains were removed thoroughly. About 18 N-terminal residues were kept intact.

Reg4 KO mice were generated by CYAGEN, China according to described strategy in Supplementary Fig. 6a. Reg4 KO mice were identified using following

**Fig. 3 E. coli promote sensitivity to DSS-mediated colitis and accumulation of inflammatory macrophages.** The survival rate (**a**), body weight (**b**), and disease index (**c**) in *E. coli* 0160 infused CFD$^{fl/fl}$pvillin-cre$^W$ (WT/0160) and CFD$^{fl/fl}$pvillin-cre$^T$ mice (CFDKO/0160) after DSS (2%) treatment (male, *n* = 15). **d** The length of colon in *E. coli* 0160 infused CFD$^{fl/fl}$pvillin-cre$^W$ (WT/0160) and CFD$^{fl/fl}$pvillin-cre$^T$ mice (CFDKO/0160) after DSS (2%) treatment (*n* = 6). **e** H&E staining of colon tissues in *E. coli* 0160 infused CFD$^{fl/fl}$pvillin-cre$^W$ (WT/0160) and CFD$^{fl/fl}$pvillin-cre$^T$ mice (CFDKO/0160) after DSS (2%) treatment. One representative from six mice. Scale bar, 40 μM. **f** QRT-PCR of inflammatory cytokines TNFα, IL-6 and IL-1β in colon tissues of *E. coli* 0160 infused CFD$^{fl/fl}$pvillin-cre$^W$ (WT/0160) and CFD$^{fl/fl}$pvillin-cre$^T$ mice (CFDKO/0160) after DSS (2%) treatment (*n* = 6). **g** Flow cytometry and number of CD45$^+$CD11b$^+$CD64$^+$CD103$^-$MHC$^+$Ly6C$^+$ and F4/80$^+$TNFα$^+$ cells in colon tissues of DSS-treated CFD$^{fl/fl}$pvillin-cre$^W$ and CFD$^{fl/fl}$pvillin-cre$^T$ mice with (Wt/*E.coli* or CFDKO/*E.coli*) or without *E. coli* colonization (Wt or CFDKO). **h** Flow cytometry of CD45$^+$CD4$^+$IFNγ and CD45$^+$CD4$^-$IFNγ$^+$NKp46$^+$ cells in DSS-treated CFD$^{fl/fl}$pvillin-cre$^T$ (CFDKO) and CFD$^{fl/fl}$pvillin-cre$^W$ (WT) mice with (Wt/*E.coli* or CFDKO/*E.coli*) or without (WT or CFDKO) *E. coli* colonization. Kruskal–Wallis test in **a**; Analysis of variance test in **b** and **c**; Two side student's *t* test in **d–h**; *P < 0.05; **P < 0.01; ***P < 0.001; NS no significance. R.E relative expression.

primers: Forward primer (F): 5′-CATCACCGTGGAAGACCTTTGC-3′ and Reverse primer (R): 5′-TTTGATTATTGGAGATAGGGTACAG-3′.

All experimental litters were bred and maintained under specific pathogen-free conditions in Animal Center of Nankai University. Experiments were carried out using age- and gender-matched mice or cohoused CFD$^{fl/fl}$pvillin-cre$^W$mice. All procedures were conducted according to the Institutional Animal Care and Use Committee of the Model Animal Research Center. Animal experiments were approved by the Institute's Animal Ethics Committee of Nankai University. All experimental variables such as husbandry, parental genotypes and environmental influences were carefully controlled.

**Mouse models.** For dextran sodium sulfate (DSS) induced colitis, DSS induced colitis was performed according to our previously reported method[56] with modification. Briefly, mice received 2.5% DSS (40,000 kDa; MP Biomedicals) or at the indicated dose in their drinking water for 7 days, then switched to regular drinking water. The amount of DSS water drank per animal was recorded and no differences in intake between strains were observed. For survival studies, mice were followed for 12 days post start of DSS-treatment. Mice were weighed every other day for the determination of percent weight change. This was calculated as: % weight change = (weight at day X-day 0/weight at day 0) × 100. Animals were also monitored clinically for rectal bleeding, diarrhea, and general signs of morbidity, including hunched posture and failure to groom. For microbiota transplantation, germ-free (GF) mice were orally administered 200 μl 1 × 10$^9$ bacteria (once/week). In *wt* mice, mice were first treated with pan-antibiotics (ampicillin (A, 1 g/L, Sigma), vancomycine (V, 0.5 g/L, Sigma), neomycin sulfate (N, 1 g/L, Sigma), and metronidazole (M, 1 g/L, Sigma)) via the drinking water for one week (sometime longer than one week), which could eliminate gut bacteria, and then orally administered 200 μl of 1 × 10$^9$ bacteria (once/week). To confirm the elimination of bacteria, stools were collected from antibiotic-treated and untreated mice and cultured in anaerobic and aerobic condition. Mice were sacrificed at the indicated days. Representative colon tissues were embedded in paraffin for hematoxylin/ eosin (H&E) staining or embedded in OCT compound (Tissue-Tek, Sakura, Torrance, CA) and frozen over liquid nitrogen for immuno-staining.

Disease activity index (DAI) and histological scores were assessed according to following method[57]. Disease activity index was the average of these scores: (combined score of stool consistency, bleeding and weight loss)/3. Diarrhea was scored daily as follows: 0, normal; 2, loose stools; 4, watery diarrhea. Blood in stool was scored as follows: 0, normal; 2, slight bleeding; 4, gross bleeding. Weight loss was scored as follows: 0, none; 1, 1–5%; 2, 5–10%; 3, 10–15%; 4, >15%. Disease activity index was the average of these scores: (combined score of stool consistency, bleeding and weight loss)/3. For histological evaluation, histology was scored as follows: epithelium (E), 0 = normal morphology; 1 = loss of goblet cells; 2 = loss of goblet cells in large areas; 3 = loss of crypts; 4 = loss of crypts in large areas; and infiltration (I), 0 = no infiltrate; 1 = infiltrate around the crypt basis; 2 = infiltrate reaching the lamina (L) muscularis mucosae; 3 = extensive infiltration reaching the L muscularis mucosae and thickening of the mucosa with abundant oedema; 4 = infiltration of the L submucosa. Total histological score was given as E + I[57].

**Gut microbiota analyses.** For gut microbiota analyses, previously reported method was used[58]. Gut microbiota was analyzed by Majorbio Biotechnology Company (Shanghai, China) using primers that target to the V3–V4 regions of 16S rRNA. Once PCR for each sample was completed, the amplicons were purified using the QIAquick PCR purification kit (Qiagen Valencia, CA), quantified, normalized, and then pooled in preparation for emulsion PCR followed by sequencing using Titanium chemistry (Roche, Basel Switzerland) according to the manufacturer's protocol. Operational Taxonomic Unit (OTU) analysis was performed as follows: sequences were processed (trimmed) using the Mothur software and subsequently clustered at 97% sequence identity using cd-hit to generate OTUs. The OTU memberships of the sequences were used to construct a sample-OTU count matrix. The samples were clustered at genus and OTU levels using the sample-genus and sample-OTU count matrices respectively. For each clustering,

Morisita-Horn dissimilarity was used to compute a sample distance matrix from the initial count matrix, and the distance matrix was subsequently used to generate a hierarchical clustering using Ward's minimum variance method. The Wilcoxon Rank Sum test was used to identify OTUs that had differential abundance in the different sample groups.

For the absolute numbers of gut bacteria, 16s rRNAs were extracted using bacterial DNA extraction kit (CWBio), including that nuclear lysis buffer split cell nuclear, protein precipitation solution precipitates and removes protein, and purified genomic DNA is precipitated by isopropanol; and then amplified using phylum such as *Firmicutes*, *Bacteroidetes*, genus or strain-specific primers. All qPCR assays were performed on CFX Connect system (BIO-RAD) in 25 μl reaction volume containing 1×SYBR Green PCR Mastermix. For each primer, a concentration of 1μM was added. The following thermal program was applied: a single cycle of DNA polymerase activation for 5 min at 95 °C followed by 40 amplification cycles of 15-s denaturing step (95 °C), 30-s annealing (55 °C) and 30-s extension step (72 °C). Afterward, melting temperature analyses of the obtained amplification products was performed using standard machine settings. The concentration of each product was detected and then exchanged into copy numbers. Primers used were listed in Supplementary Table 1.

For compositional analysis of the intestinal tract and extra intestinal tissues, previously reported method was used[6]. Briefly, whole tissues were harvested and homogenized them using a Polytron PT2100 homogenizer at 17000 rpm (Kinematica) in sterile thioglycolate media. We then serially diluted the homogenate and plated on LB agar and Schaedler agar supplemented with 5% defibrinated sheep blood plates and grew at 37 °C aerobically and anaerobically for 24 h. We counted bacterial colonies and then separated based on colony appearance and performed colony 16S rDNA PCR and sequencing.

For colony PCR, cecal contents from mice were diluted and plated onto LB agar. The colonies from each animal were picked, resuspended in sterile water, and heated to 100 °C for 15 min to lyse bacteria. PCR was performed and PCR products were resolved on a 1% agarose gel.

**Generation of GFP-labeled E. coli.** Competent cells (5 × 10$^6$) were prepared and transfected using 500 ng pEGFP plasmids according to supper-competent cell preparation kit (85% LB medium, 10% PEG, 5% DMSO and 50 mM MgCl$_2$, Beyotime) and then selected using ampicillin.

**Cell isolation and flow cytometry.** For the staining of lamina propria (LP) lymphocytes, previously reported method[59] was used. Briefly, colon or small intestine were isolated, cleaned by shaking in ice-cold PBS four times before tissue was cut into 1 cm pieces. The epithelial cells were removed by incubating the tissue in HBSS with 2 mM EDTA for 30 min at 37 °C with shaking. The LP cells were isolated by incubating the tissues in digestion buffer (DMEM, 5% fetal bovine serum, 1 mg/ ml Collagenase IV (Sigma-Aldrich) and DNase I (Sigma-Aldrich) for 40 min. The digested tissues were then filtered through a 40-mm filter. Cells were resuspended in 10 ml of the 40% fraction of a 40: 80 Percoll gradient and overlaid on 5 ml of the 80% fraction in a 15 ml Falcon tube. Percoll gradient separation was performed by centrifugation for 20 min at 1800 rpm at room temperature. LP cells were collected at the interphase of the Percoll gradient, washed and resuspended in medium, and then stained and analyzed by flow cytometry. Dead cells were eliminated through 7-AAD staining.

For analysis of different immune cell populations, previously reported method[60] was used. Briefly, the cells were washed with staining buffer containing 2% FBS, 1 mM EDTA and 0.09% NaN3 and surface staining was performed with APC, FITC, PercP, BV 605 or PE-labeled anti-CD4, CD8, NKp46, CD11c, MHCII, F4/80, CD11b, Ly6C and CD45 antibodies and analyzed using FACScan flow cytometry.

For intracellular staining, the cells were cultured and stimulated for 6 h with 50 ng/ml phorbol 12-myristate 13-acetate (PMA, Sigma) and 1 μg/ml ionomycin (Sigma) in the presence of GolgiStop (10 ng/ml, BD Biosciences). After incubation

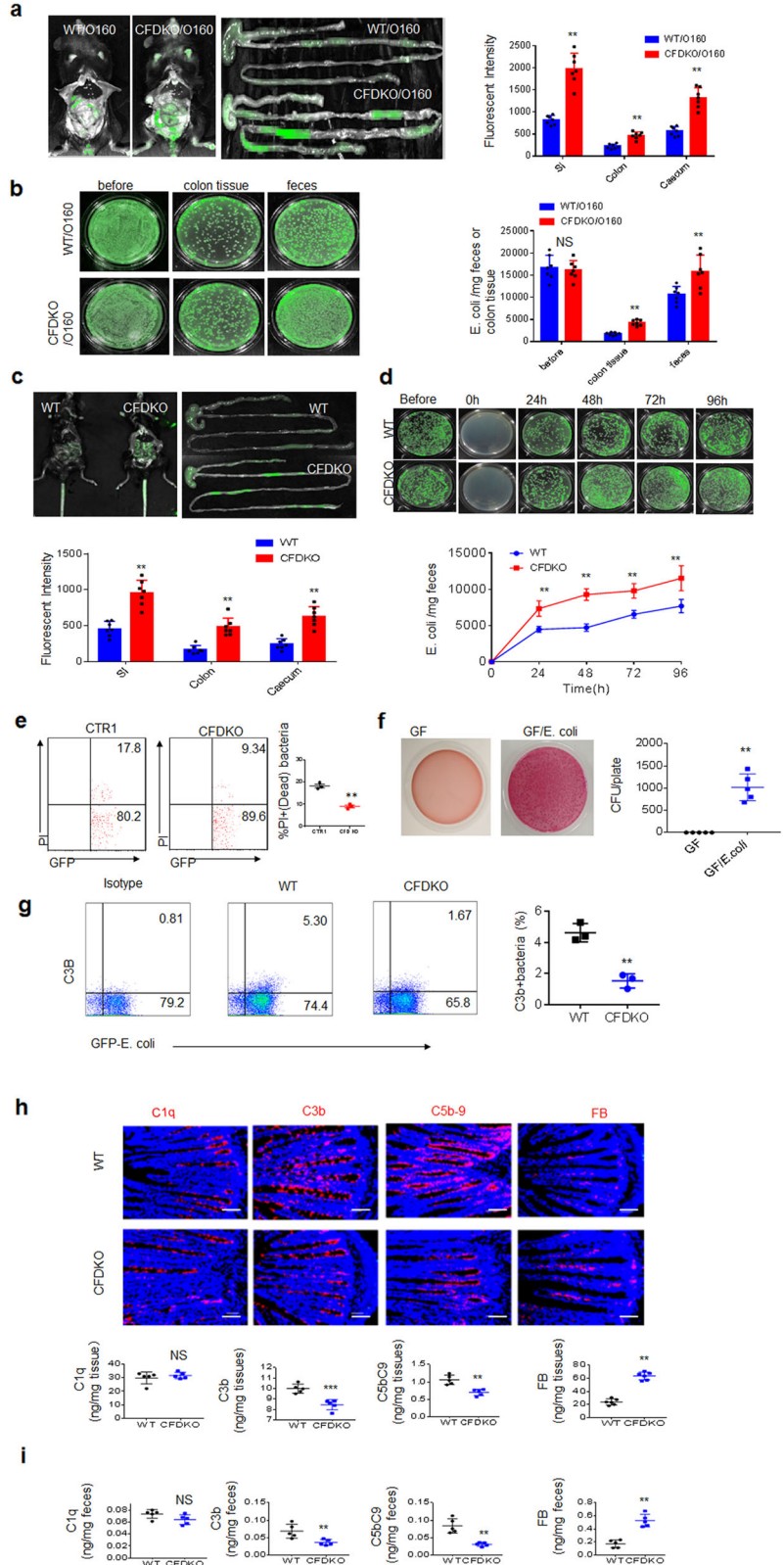

for 6 h, cells were washed in PBS, and then fixed in Cytofix/Cytoperm, permeabilized with Perm/Wash buffer (BD Biosciences), and stained with FITC-, PE-, APC- APC/cy7-, PerCP/Cy5.5- or PE/cy7-conjugated antibodies. Meanwhile, dead cells were eliminated through 7-AAD staining.

For analyses of C3b-, Reg4-, IgA- or collectin-coating of *E. coli* in colonic contents, fresh colonic contents was resuspended in sterile PBS. An aliquot estimated to contain no more that GFP-labeled $10^6$ *E. coli* was directly stained with

a monoclonal body. After washing, secondary antibodies were added. After a final washing step, samples were analyzed on the BD flow cytometry with setting adapted for optimal detection of bacterial-sized particle.

**Serum sensitivity assays**. For serum sensitivity assays, previously reported method[61] was used. Briefly, bacteria were then scraped off the plates and suspended

**Fig. 4 CFD<sup>fl/fl</sup>pvillin-cre<sup>T</sup> mice have reduced bactericidal abilities against *E. coli*. a** Fluorescence intensity in small intestine (SI), colon and caecum of CFD<sup>fl/fl</sup>pvillin-cre<sup>W</sup> (WT/0160) and CFD<sup>fl/fl</sup>pvillin-cre<sup>T</sup> mice (CFDKO/0160) after infusing GFP-labeled *E. coli* for 48 h (*n* = 6). **b** *E.coli* clones in the colon tissues and feces of CFD<sup>fl/fl</sup>pvillin-cre<sup>W</sup> (WT/0160) and CFD<sup>fl/fl</sup>pvillin-cre<sup>T</sup> mice (CFDKO/0160) after infusing GFP-labeled *E. coli* for 48 h (*n* = 6). **c** Fluorescence intensity in small intestine (SI), colon and caecum of CFD<sup>fl/fl</sup>pvillin-cre<sup>W</sup> (WT/0160) and CFD<sup>fl/fl</sup>pvillin-cre<sup>T</sup> (CFD KO) germ-free (GF) mice after infusing GFP-labeled *E. coli* for 48 h (*n* = 6). **d** *E. coli* clones in the feces of GFP-labeled *E. coli* infused CFD<sup>fl/fl</sup>pvillin-cre<sup>W</sup> (WT/0160) and CFD<sup>fl/fl</sup>pvillin-cre<sup>T</sup> (CFDKO/0160) GF mice at the indicated time (*n* = 6). **e** Flow cytometry of live (GFP + PI−)/dead(GFP + PI+) bacteria. The equal amount of GFP-labeled *E.coli* (E. coli 0160,1 × 10$^7$) were infused into different mice, including CTR1 (CFD<sup>fl/fl</sup>pvillin-cre<sup>W</sup>mice) and CFDKO(CFD<sup>fl/fl</sup>pvillin-cre<sup>T</sup>mice). After 48 h, the feces from different mice were analyzed using flow cytometry for live (GFP + PI−)/dead(GFP + PI+) bacteria. **f** GF mice were infused by *E. coli* (GF/E. coli) and then feces were measured for CFU of bacteria after one week. **g** Flow cytometry of C3b deposited on GFP-labeled *E. coli* in GFP-labeled *E. coli* infused CFD<sup>fl/fl</sup>pvillin-cre<sup>W</sup> (WT) and CFD<sup>fl/fl</sup>pvillin-cre<sup>T</sup> (CFD) mice (*n* = 3). **h** Immuo-fluorecence staining (upper) and ELISA (lower) of C1q, C3, C5b-9 and FB in the colon tissues of no-treated CFD<sup>fl/fl</sup>pvillin-cre<sup>W</sup> (WT) and CFD<sup>fl/fl</sup>pvillin-cre<sup>T</sup> (CFDKO) mice (*n* = 5). Scale bar, 40 μM. **i** ELISA of C1q, C3, C5b-9 and FB in the feces of no-treated CFD<sup>fl/fl</sup>pvillin-cre<sup>W</sup> (WT) and CFD<sup>fl/fl</sup>pvillin-cre<sup>T</sup> (CFDKO) mice (*n* = 5). Two side student's *t* test in **a–c**, **e–i**; Analysis of variance test in **d**. *$P < 0.05$; **$P < 0.01$; ***$P < 0.001$; NS no significance.

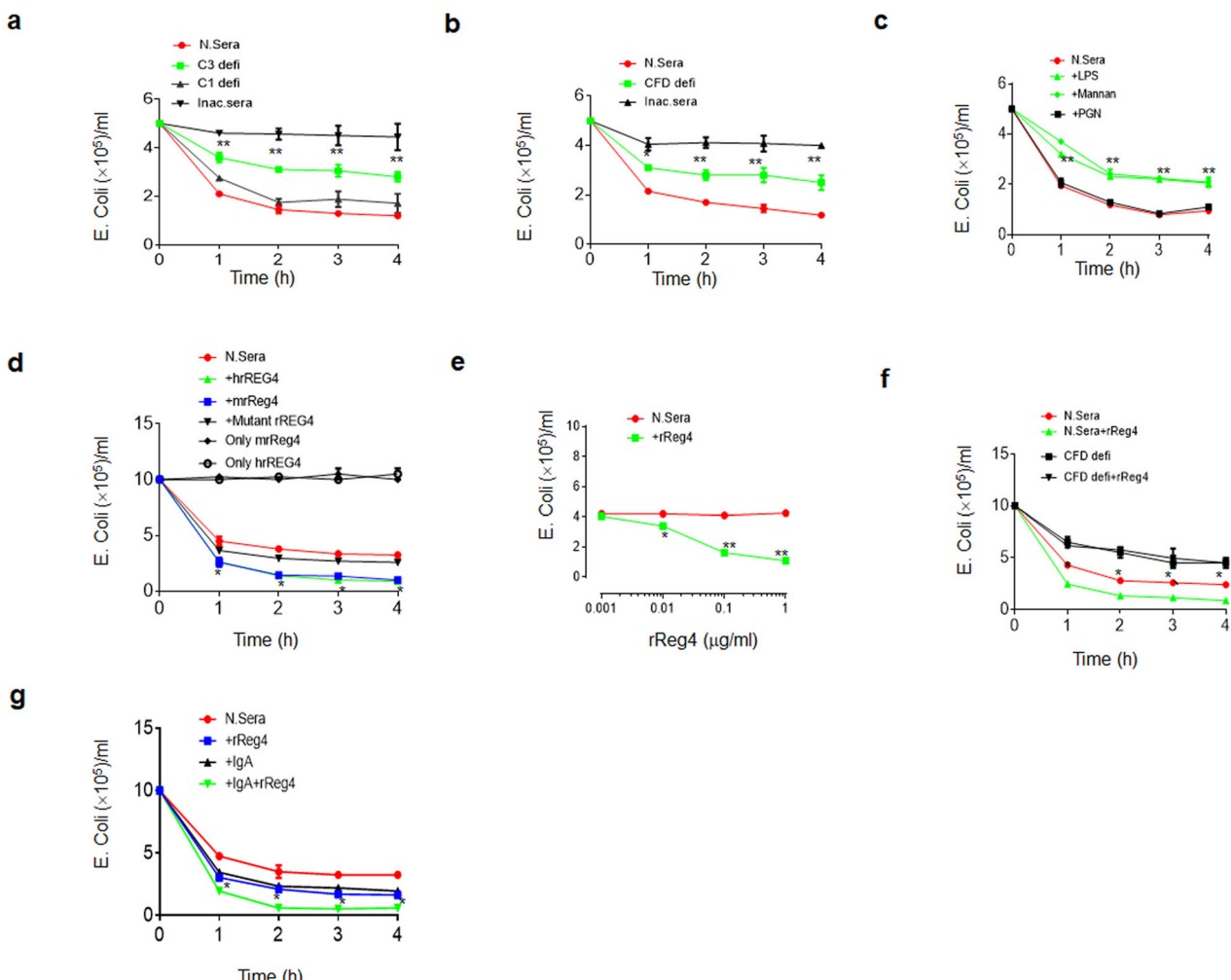

**Fig. 5 Reg4 is involved in the bactericidal activity to *E. coli*. a** *E. coli* clones in the normal human sera (N. Sera, Quidel), C3deficient sera (C3defi, Quidel), C1 deficient sera (C1 defi, Quidel) and inactive sera at different time points. *C3deficint sera vs normal sera. **b** *E. coli* clones in the normal sera (N. Sera, Quidel), CFD-deficient sera (CFD defi, Quidel) inactive sera at different time points. *CFD-deficient sera vs normal sera. **c** *E. coli* clones in the normal human sera after adding LPS (containing mannose), mannose or peptidoglycan. *LPS vs normal sera. **d** *E. coli* clones in the normal human sera after adding rReg4, rREG4 or mutated rREG4. *Reg4 vs normal sera. Only mrReg4 and only hrREG4, *E.coli* clones in inactive complement serum after adding mrReg4 or hrREG4. **e** *E. coli* clones in the normal human sera after adding different concentrations of rReg4. **f** *E. coli* clones in the normal human sera (N. sera) or CFD-deficient (CFD defi) sera after adding rReg4 (N. sera/rReg4 or CFDdefi/rReg4). *Normal sera/Reg4 vs normal sera. **g** *E. coli* clones in the normal human sera after adding rREG4, IgA or IgA+REG4. *IgA+Reg4 vs Reg4. Around 5 × 10$^5$ in **a–c** or 1 × 10$^6$ in **d–g** E. coli CFUs were added into 1 ml 10% sera. Bacterial suspension were diluted and plated on MacConkey agar plates. Plates were incubated overnight at 37 °C, and CFU were counted. Analysis of variance test used in **a–g**. *$P < 0.05$; **$P < 0.01$; ***$P < 0.001$; NS no significance. One representative of three independent experiments.

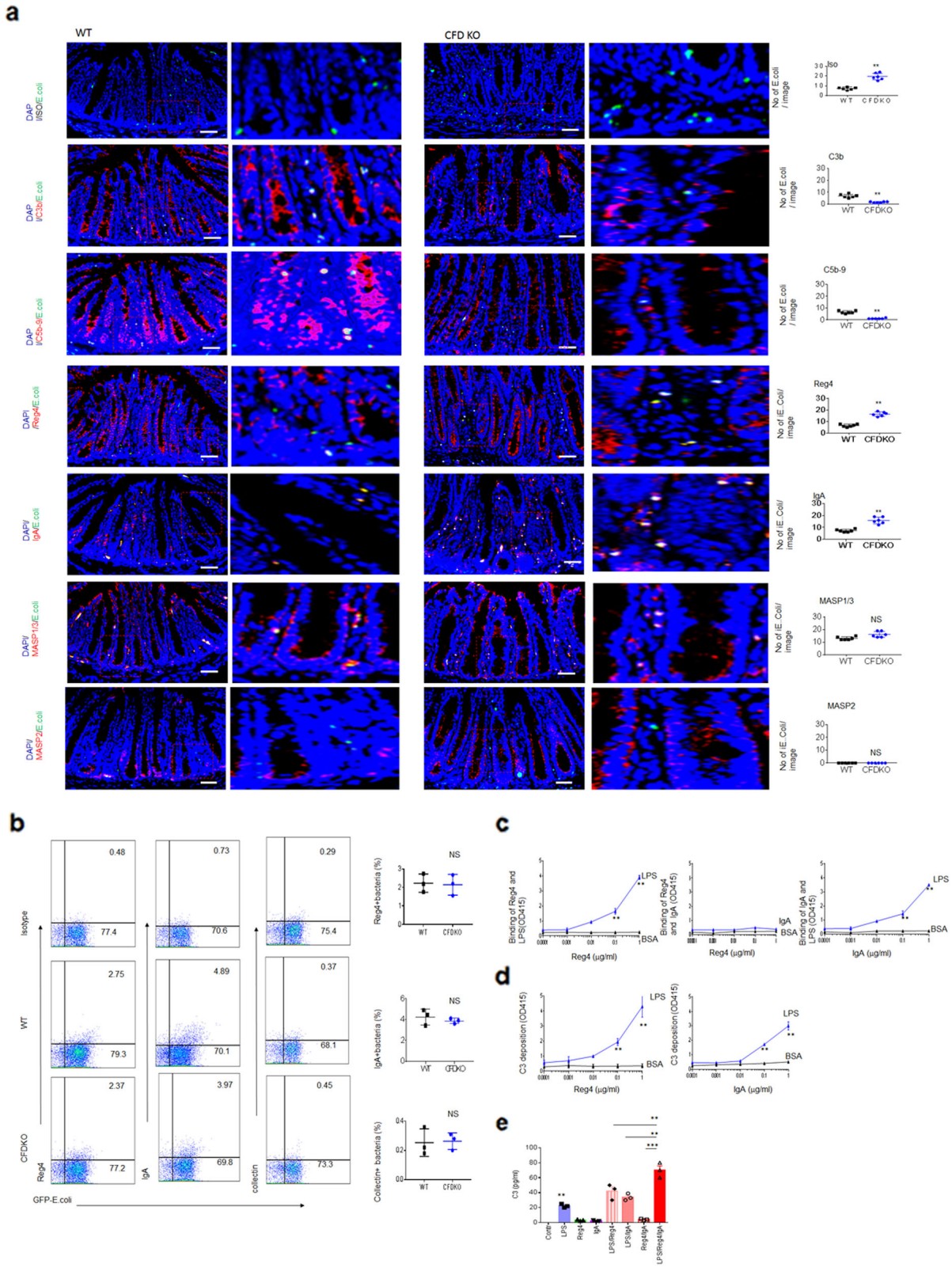

in phosphate-buffered saline (PBS). Optical density of the solutions was measured at 600 nm using a spectrophotometer (Thermo Scientific, Waltham, MA, USA) and adjusted to an optical density of 0.700. $5 \times 10^5$ or $1 \times 10^6$ CFU was added into PBS that contains 10% Normal Human Complement Standard (Quidel), C1q-depleted serum (Quidel) or C3 depleted and CFD depleted serum (Quidel). Mixtures were

incubated at 37 °C and sampled at 0, 1, 2, 3 and 4 h. Bacterial suspensions were diluted 1:100 in PBS and plated on LB agar plates. Plates were incubated overnight at 37 °C. Resultant bacterial CFU were counted, and CFU per mL of bacterial suspension were calculated. All serum sensitivity assays were repeated at least three times.

**Fig. 6 Gut reg4/IgA initiate membrane attack complex on E.coli. a** Immunostaining of C3b, C5-9, Reg4, IgA, MASP1/3 and MASP2 on GFP-labeled E. coli in 2% DSS-treated GFP-labeled E. coli infused CFD$^{fl/fl}$pvillin-cre$^T$ (CFDKO/0160) mice (Left) and statistic analyses (right, three images/slide, three slides/mouse, $n = 6$). Scale bar, 40 μM. No, number; Iso, isotypic antibod. **b** Flow cytometry of Reg4, IgA, MASP1/3 and MASP2 on the GFP-labeled E. coli in untreated GFP-labeled E. coli infused CFD$^{fl/fl}$pvillin-cre$^w$ (WT) and CFD$^{fl/fl}$pvillin-cre$^T$ (CFDKO) mice and statistic analyses ($n = 3$). **c** ELISA for detecting binding of Reg4 with LPS (left), Reg4 with IgA (middle) or IgA with LPS (right) in different concentration of Reg4 (left and middle) or IgA (right) coated plates **d** ELISA of C3 in different concentration of Reg4 (left) or IgA (right) with LPS coated plates. For complement resources, 5% normal human sera were added. **e** ELISA of C3b in the LPS, Reg4, IgA, LPS + Reg4 (LPS/Reg4), LPS + IgA (LPS/IgA), Reg4+IgA (Reg4/IgA) and LPS + Reg4+IgA (LPS/Reg4/IgA) coated plates. For complement resources, 5% normal human sera were added. Mann–Whitney U test in **a**; Two side student's t test in **b**; ANOVA plus post-Bonferroni analysis in **e**; Analysis of variance test in **c** and **d**. *$P < 0.05$; **$P < 0.01$; ***$P < 0.001$; NS no significance. In **c–e**, one representative of three independent experiments.

**Blocking experiments**. For blocking experiments, LPS (100 mM) or mannan (100 mM) were added into PBS that contains 10% Normal Human Complement Standard (Quidel); Recombinant mouse Reg4 (rReg4, 100 ng/ml), human recombinant REG4 (100 ng/ml), mutant rREG4 (100 ng/ml) and IgA (100 ng/ml) were also respectively added into PBS that containing 10% Normal Human Complement Standard (Quidel). The tubes were further incubated at 37 °C in a $CO_2$ incubator using a rotator (multitube tube with a fixed speed; Thermo Fisher) for 1–4 h, at which equal amount of liquids were removed and plated onto chocolate agar to quantify surviving CFUs per milliliter. To ascertain input CFUs per milliliter, aliquots were also removed at time 0 from control tubes containing buffer alone or the subjects' plasma that had been heated to 56 °C for 30 min to inactivate complement.

**MAC formation**. For MAC formation, LPS (1 μg/ml), IgA (10 μg/ml), Reg4 (10 μg/ml), LPS + IgA (10 μg/ml + 1 μg/ml), LPS + Reg4 (100 ng/ml + 1 μg/ml) or LPS + Reg4 + IgA (100 ng/ml + 1 μg/ml + 1 μg/ml) were coated on the 96 well plates, and then contains 5% Normal Human Complement Standard were added as complement resources. MAC formation was determined by anti-C3 ELISA. For this assay, all washing and incubation steps were performed in the absence of Tween 20, which reduced nonspecific staining. For detecting binding of Reg4 with LPS (left), Reg4 with IgA (middle) or IgA with LPS (right), different concentrations of Reg4 or IgA were coated on the plates and then detected by ELISA.

For C1q, BP, C5b-9 and C3 assays, tissues and colonic content samples were detected for C1q, BP, C5b-9 and C3 by ELISA (Quidel Corporation, San Diego, CA, USA) respectively and according to the manufacturer's instruction.

**Generation of mouse Reg4, human REG4 and mutant human REG4**. To generated the mouse Reg4 and human REG4, the DNA sequence encoding mouse Reg4 and human REG4 with an N-terminal His-tag was cloned into the pEASY®-Blunt E2 Expression vector (Transgen) and transformed into E.coli BL21. The site-directed mutant human REG4 Δ P 91S was prepared by use of the Fast MultiSite Mutagenesis System (Transgen), and the coding regions were confirmed by DNA sequencing. The protein samples were expressed after stimulation with 1 mM IPTG in LB medium, and then purified using ProteinIso® Ni-NTA Resin (Transgen).

**H & E staining**. For hematoxylin/eosin (H&E) staining, previously reported methods were used in this experiment[60]. Briefly, the entire colon was excised to measure the length of the colon and then were fixed in 4% (w/v) paraformaldehyde buffered saline and embedded in paraffin, 5 μm sections colon sections were cut and stained with H&E.

**Immunostaining**. Immunostaining was performed according to reported method[58]. Briefly, colon tissues were embedded in OCT compound (Tissue-Tek, Torrance, CA) and frozen over liquid nitrogen. 5-μm-thick sections were prepared from frozen tissue and fixed in acetone (−20 °C) for 10 min. After rehydration in PBS for 5 min and further washing in PBS, tissue sections were blocked with 1% (w/v) BSA and 0.2% (w/v) milk powder in PBS (PBS-BB). The primary antibody was added in PBS-BB and incubated overnight at 4 °C. After washing (three times, 5 min each), tissue was detected with DAB kit or fluorescence labeled second antibody. Nuclei were stained by DAPI.

**Fluorescent in situ hybridization**. For fluorescent in situ hybridization (FISH), mucus immune-staining was paired with fluorescent in situ hybridization (FISH) in order to analyze bacteria localization at the surface of the intestinal mucosa according to reported method[58]. In brief, the ileum and colonic tissues (proximal colon, second centimeters from the caecum) containing fecal material were placed in methanol-Carnoy's fixative solution (60% methanol, 30% chloroform, 10% glacial acetic acid) for a minimum of 3 h at room temperature. Tissue were then washed in methanol, ethanol, ethanol/xylene (1:1) and xylene, followed by embedding in paraffin with a vertical orientation. 5-μm sections were cut and dewaxed by preheating at 60 °C for 10 min, followed by bathing in xylene at 60 °C

for 10 min, xylene at room temperature for 10 min and 99.5% ethanol for 10 min. The hybridization step was performed at 50 °C overnight with an probe diluted to a final concentration of 0.01 μg/mL in hybridization buffer (20 mM Tris-HCl, pH 7.4, 0.9 M NaCl, 0.1% SDS, 20% formamide). After washing for 10 min in wash buffer (20 mM Tris-HCl, pH 7.4, 0.9 M NaCl) and 10 min in PBS and block solution (5% FBS in PBS) was added for 30 min at 50 °C. Mucin 2 primary antibody (rabbit H-300, Santa) was diluted to 1: 200 in block solution and applied overnight at 4 °C. After washing in PBS, block solution containing anti-rabbit secondary antibody diluted to 1:200 was applied to the section for 2 h. Nuclei were stained using Hoechst33342.

**Immunoblot**. Immunoprecipitation and immunoblot were performed according to previous methods[26]. The cells were lysed with cell lysis buffer (Cell Signaling Technology), which was supplemented with a protease inhibitor 'cocktail' (Calbiochem). The protein concentrations of the extracts were measured using a bicinchoninic acid assay (Pierce). For the immunoblot, hybridizations with primary antibodies were conducted for 1 h at room temperature in blocking buffer. The protein-antibody complexes were detected using peroxidase-conjugated secondary antibodies (Boehringer Mannheim) and enhanced chemiluminescence (Amersham).

**RT-PCR and qRT-PCR**. RT-PCR and qRT-PCR were performed according to our previous methods[58]. Total RNA was extracted from the cells, tissues and organs using TRIzol reagent (Invitrogen). First-strand cDNA was generated from total RNA using oligo-dT primers and reverse transcriptase (Invitrogen Corp). The PCR products were visualized on 1.0% (wt/vol) agarose gels. Quantitative real-time PCR (qRT-PCR) was conducted using QuantiTect SYBR Green PCR Master Mix (Qiagen) and specific primers in an ABI Prism 7000 analyzer (Applied Biosystems). GAPDH mRNA expression was detected in each experimental sample as an endogenous control. All reactions were run in triplicate. The primers used in this study were listed in Supplementary Data 1 of the Methods.

**Ex vivo stimulation**. For ex vivo colon stimulation, colon from healthy mice were harvested, washed and incubated with or without $1 \times 10^9$ E. coli in DMEM media with ATP (2 mM) for 1 h, For analyses of IL-18, the colon epithelial cells were separated from colon tissues using 0.1% EDTA, followed by three 1 min shakings by hand, a 15-min incubation at 4 °C, and passage through 70-μm filters (BD Falcon) to collect the flow through. Fraction containing intact and isolated crypts were collected by centrifugation at $75 \times g$ for 5 min. at 4 °C and washed with PBS. The lamina propria was separated from crypts to enrich for mononuclear and intestinal epithelial cells, respectively. Protein extracts were analyzed by immunoblotting for mature forms of IL-18. The supernatants were collected for IL-18 ELISA.

For caspase inhibition experiment, pan-caspase (100 μM), caspase 1(100 μM) and caspase 8 inhibitor (46 μM) were respectively added into culture, and then colon epithelial cells were separated, and expression of IL-18 was analyzed using immunoblotting and ELISA.

For PKCδ inhibition experiment, PKCδ inhibitors (20 μM) were added into culture, and then colon epithelial cells were separated at the indicated time. The expression of IL-18 was analyzed using immunoblotting and ELISA.

**Statistics and reproducibility**. Student's t test and ONE-way ANOVA Bonferroni's Multiple Comparison Test was used to determine significance. The statistical significance of the survival curves was estimated using Kaplan and Meier method, and the curves were compared using the generalized Wilcoxon's test. Histological scores in different groups were analyzed by a Mann–Whitney U test. A 95% confidence interval was considered significant and was defined as $p < 0.05$.
* indicates $p < 0.05$, **$p < 0.01$, ***$p < 0.001$.

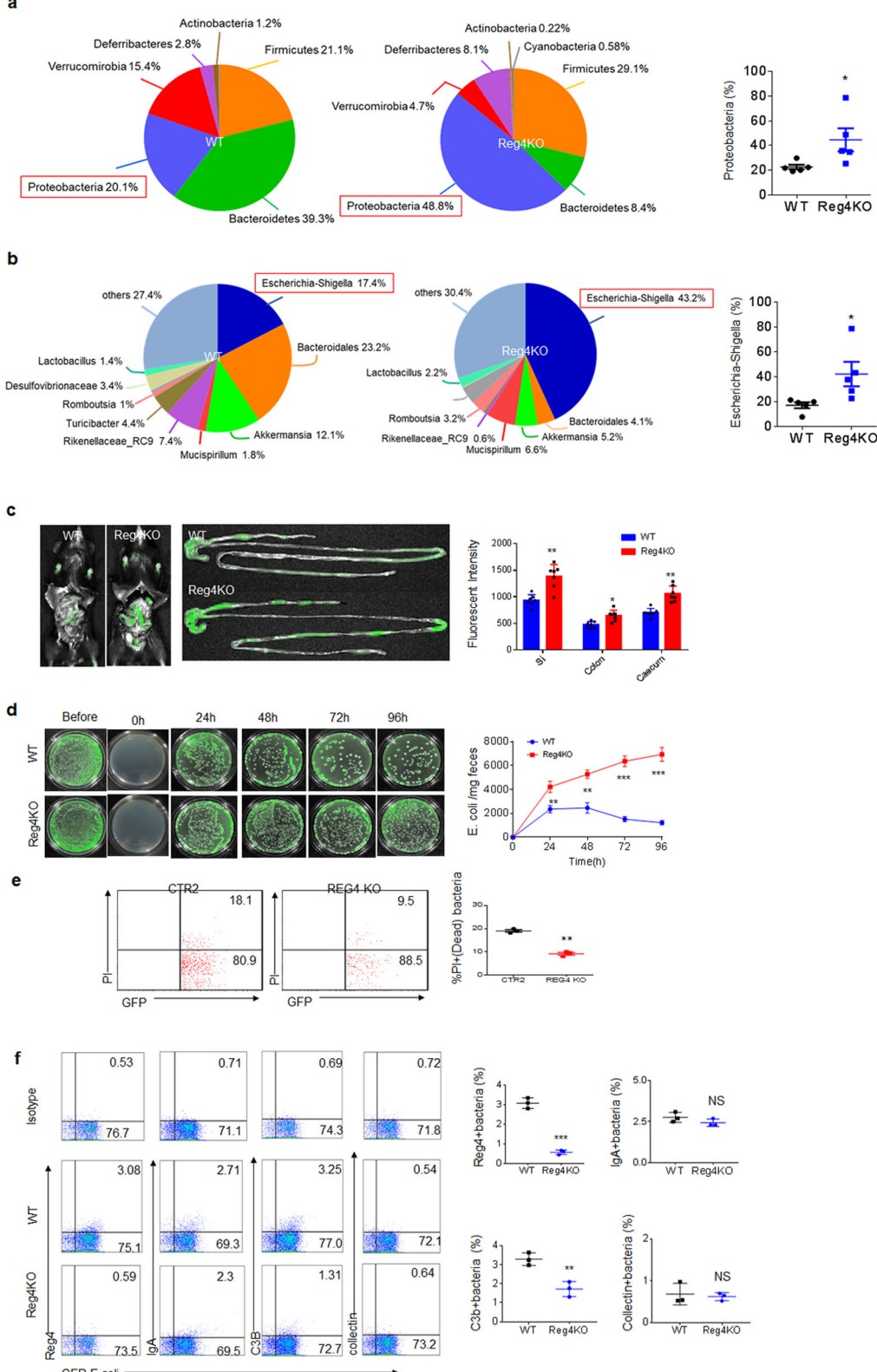

**Fig. 7 Reg 4 deficient mice have reduced bactericidal abilities against *E. coli*. a** and **b** The proportion of bacteria after 16s rRNA analyses of colon contents in *wt* and Reg4 deficient mice after 2% DSS ($n = 5$). Also, see Fig. S6B. **c** Fluorescence intensity in small intestine (SI), colon and caecum after infusing *wt* and Reg4 KO using GFP-labeled *E. coli* for 48 h ($n = 6$). **d** *E.coli* clones in the feces of WT and Reg4 KO after infusing GFP-labeled *E. coli* at the indicated time ($n = 6$). **e** Flow cytometry of live (GFP + PI−)/dead(GFP + PI+) bacteria. The equal amount of GFP-labeled *E.coli* (*E. coli* 0160, $1 \times 10^7$) were infused into different mice, including CTR2 (WT) and REG4 KO mice. After 48 h, the feces from different mice were analyzed using flow cytometry for live (GFP + PI−)/dead(GFP + PI+) bacteria. **f** Flow cytometry of Reg4, IgA, C3b and collectin deposited *E. coli* and statistic analyses ($n = 3$). Two side student's *t* test in **c**, **e** and **f**; Analysis of variance test in **d**. *$P < 0.05$; **$P < 0.01$; ***$P < 0.001$; NS no significance.

**Reporting summary**. Further information on research design is available in the Nature Research Reporting Summary linked to this article.

## Data availability

16S rRNA gene sequence data include:16S rRNA gene sequence data of colon feces derived from WT and CFDKO mice. PRJNA577385; 2) 16S rRNA gene sequence data of colon feces derived from DSS-treated WT and normal WT mice. PRJNA577387; 16S rRNA gene sequence data of colon feces derived from WT and CFDKO mice after DSS treatment. PRJNA577388; 16S rRNA gene sequence data of colon feces derived from wild type and Reg4 KO mice after DSS treatment. PRJNA577541. Full blots are shown in Supplementary Information. Source data underlying plots shown in figures are provided in Supplementary Data 2. All other data (if any) are available upon reasonable request.

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

## Acknowledgements

This research was supported by NSFC grants 91842302, 81970488, 91029736, 9162910, 91442111 and 31570114; National Key Research and Development Program of China (2016YFC1303604), Tianjin Municipal Science and Technology Bureau (18JCZDJC35300), The State Key Laboratory of Medicinal Chemical Biology, "the Fundamental Research Funds for the Central Universities", Nankai University (Grant Number 63191724).

## Author contributions

R.Y. designed the research and wrote the paper; H.Q., J.W., Y.G., conducted in vivo experiments and immunoassay, participated in study design and performed the statistical analysis; Y.Y., Y.L., X.S., performed partly in vivo assay; H.Z., L.S., prepared germ-free mice; Y.Z. offered assistance for the animal experiments. All authors read and approved the final manuscript.

## Competing interests

The authors declare no competing interests.
