## [Peer Review File · Communications Biology]

Reviewers' comments:

Reviewer #1 (Remarks to the Author):

In this manuscript, Qi et al demonstrated that both gut epithelial cell-derived CFD and Reg4 play important roles in eliminating E. coli and thus maintaining gut homeostasis, through mechanisms mainly involving the complement-mediated attach complexes. In general, the topic is novel, interesting and important. However, the results presented in this manuscript are loosely connected with each other, and the mechanisms described are quite broad and multi-dimensional rather than deeply investigated. There are two major issues that the authors need to be concerned about:

1. What is the relationship between CFD and Reg4? The authors seem to separate them into 2 different stories in the manuscript, although both of which are claimed to be associated with complement-mediated activity. The only evidence showing the relationship between CFD and Reg4 is that Reg4/REG4 promotes the killing of E.coli in normal serum but not CFD deficient sera in vitro (Figure 5f), so it is insufficient and not convincing to conclude that "We also found that gut epithelia cells derived CFD play a critical role in Reg4-mediated bactericidal activity on the E. coli" (line 283 and 284). How about the activity of CFD in Reg4 deficient mice and vice versa? Does the deletion of CFD in Reg4 deficient mice influence the phenotypes and vice versa? These are important questions yet to be answered.
2. The authors tried to explain the DSS and E.coli phenotypes in CFDKO vs. WT mice through different mechanisms involving inflammatory macrophages, inflammasome-related protein production (IL18, NLRC4, caspase1, etc) and complement-mediated bactericidal activity. Can the authors clarify whether these mechanisms are related with each other and what is the key mechanism in mediating these phenotypes?

There are some minor critiques:

1. In line 108, 156, 160, the authors use "infusion" to describe fecal microbiome transplantation in germ-free mice, do the authors mean "colonize"? Please also provide evidence that germ-free mice are successfully colonized with mouse feces, E. coli or lactobacilli.
2. In Figure 1G, why the histological scores of some mice exceed 8 if calculated by summing up epithelium and infiltrations scores (maximum scores of each is 4 as described in the methods)? Also, the authors stated in the figure legends that there are 3 slides/mouse and n=6 mice per group, but the number of samples presented in figure1G apparently doesn't fit. Please clarify.
3. Since the DSS colitis model is strongly related with gut microbiome, is the DSS phenotype in CFDKO mice due to differences in the microbiome between KO and WT mice? The authors state that there are no remarkable changes in proportion of Proteobacteria between steady states KO and WT littermates, what about the other main genera? It is important to clarify if there is difference in overall microbiome configuration between KO and WT mice before any treatment, using parameters like alpha diversity, beta diversity and taxonomy composition.
4. In Figure 4f, g, are the C1q, C3, C5b-9 and FB levels measured in non-treated or treated CFDKO/WT mice? Please clarify.
5. The authors seem to emphasize that CFD and Reg 4 are mainly expressed in Paneth cells. However, the use of "villin-cre" mice can only demonstrate the role of CFD/ Reg 4 in all epithelial cells, not only Paneth cells. This is an acceptable limitation but needs to be mentioned.
6. There are some grammar mistakes in the manuscript. E.g, line 115 should be "was selectively knocked out"; line 136 should be "proportion"; line 217 should be "sensitive"; line 242, should be "MASP" instead of "WASP". Please check thoroughly and correct the grammar mistakes.

Reviewer #2 (Remarks to the Author):

In the manuscript "Gut epithelial cell derived Reg4 and complement factor D switch bactericidal activity on *E. coli* to maintain gut homeostasis" by Qi et al, the authors describe a protective role for complement factor D expression by intestinal epithelial cells during intestinal inflammation, in part by limiting growth of enteric *E. coli*. Intestinal epithelial-specific complement factor D conditional knockout mice (CFD^{fl/fl}-villin-cre, herein referred to as CFDKO mice as described in the manuscript) were generated and exposed to the DSS model of experimental colitis, which resulted in increased mortality and morbidity associated with elevated colonic inflammatory cytokines and lamina propria macrophages. In addition, DSS-treated CFDKO mice displayed alterations in intestinal microbiota, including increased Proteobacteria and *Escherichia-Shigella*. Transfer of *E. coli* exacerbates DSS-mediated colitis and in both SPF and germ-free CFDKO mice, accompanied by an increase in *E. coli* growth in CFDKO mice, suggesting epithelial CFD promotes bactericidal activity in vivo. CFDKO mice display reduced C3b expression and binding to *E. coli*, and serum from CFDKO mice displayed diminished capacity to restrict *E. coli* growth in vitro. Interestingly, Reg4 enhanced the bactericidal activity of wild type serum, but failed to do so in CFDKO serum, suggesting this c-type lectin requires CFD for killing of *E. coli*. Lastly, the authors show that Reg4 KO mice display a similar increased susceptibility to DSS colitis, increased *E. coli* expansion and reduced C3b deposition on *E. coli*, indicating that Reg4 functions with CFD to restrict *E. coli* via complement-mediated killing. The authors provide compelling evidence for a protective role for intestinal epithelial-derived CFD during intestinal inflammation and limiting overgrowth of *E. coli*, however additional controls and experiments will help shed mechanistic insight on the cellular contributors as well as the order of events by which these events proceed.

Major points

1. In figure 1, the authors convincingly show that CFDKO mice display increased intestinal inflammation during DSS colitis. Interestingly, untreated CFDKO mice also exhibit increased intestinal macrophages compared to naïve wild type littermates. Does this also result in elevated colonic TNF, IL6 and IL1b mRNA, similar to DSS-treated mice?
2. Similar to above, the authors show DSS-treated CFDKO mice have increased Proteobacteria and *E. coli* compared to DSS-treated wild type mice. The untreated CFDKO mice have modest (non-significant?) difference in Proteobacteria, but it is unclear from Extended Fig. 3a if they also have elevated *E. coli* prior to DSS. Were these levels detectable? This control is important for understanding if the loss of CFD limits the overgrowth of *E. coli* in naïve animals, which may contribute to the increased lamina propria macrophages during both homeostasis and intestinal inflammation. An elevation in baseline *E. coli* in CFDKO mice would also be consistent with the comparable levels of pro-inflammatory cytokines and macrophages between *E. coli*-fed wild type mice and non-colonized CFDKO mice (Fig. 3f-g).
3. In Extended Fig. 4c, the authors compare IL-18 expression in *E. coli*-fed Nlr4 and caspase-1/11 knockout mice to CFDKO mice and conclude that the increase in IL-18 observed in CFDKO mice is due to Nlr4 and caspase-1/11. However, this cannot be concluded unless the authors target Nlr4 and caspase-1/11 in the CFDKO mice, similar to the caspase-8 and PKC δ inhibitor studies shown in Extended figure 4d. This would require making double knockout animals of CFD and Nlr4 or caspase-1/11 to conclude this. However, a comparison of Nlr4 and caspase-1/11 to wild type animals would allow the authors to determine if their strain of *E. coli* activates IL-18 via Nlr4 and caspase-1/11. This could implicate the involvement of these proteins during *E. coli* recognition, but would not fully address if the increased IL-18 in the CFDKO mice is through Nlr4 and caspase-1/11.
4. The authors conclude that CFD regulates *E. coli* killing via membrane attack complex (MAC) formation. They nicely measure deposition of the early and central components that initiate the

complement cascade (i.e., C3b and MASPs), but do not measure differences in deposition of the downstream C5b-C9 components on *E. coli*, which compose the MAC.

5. Similar to above, a major function of C3b is to act as an opsonin for phagocytic cells. Given the authors demonstrate increases in intestinal macrophages in CFDKO mice, it stands to reason that CFD may also regulate *E. coli*-associated intestinal inflammation by promoting C3b-mediated phagocytosis and killing by macrophages. The authors should explore this *in vitro* by performing macrophage killing assays with *E. coli* pretreated with serum from wild type vs. CFDKO mice. This experiment would help to connect the author's striking macrophage results *in vivo* with their proposed mechanism of CFD.

Minor points

In Figure 6a, the authors show microscopy results using CFDKO, but not wild type controls. Please include this result for improved comparisons. In addition, figure 6a presents an interesting finding that CFDKO mice have increased presence of Reg4-associated *E. coli*. Is this due to an overall increase in Reg4 in the CFDKO mice? This could be addressed by the above request for wild type image comparisons.

As *E. coli* can induce intestinal inflammation on its own, do the inflammation-prone CFDKO mice display greater symptoms of disease/pathology when colonized with the 0160 strain of *E. coli* (in the absence of DSS)?

In Figure 7e, the y axis of the flow plots display "Reg4", which would indicate that some of the presented Reg4KO mice only slightly lower levels of Reg4-bound *E. coli* compared to wild type. Is this label correct as one would expect Reg4 to be absent in these mice?

Reviewer #3 (Remarks to the Author):

In the current manuscript, Qi et al. investigate how the intestinal mucosa might control the blooms of Enterobacteriaceae (e.g. *E. coli*) that have been described to occur in several instances of inflammation-associated dysbiosis. They investigate a potential role for complement-mediated bactericidal activity in this process and identify epithelial production of complement factor D (CFD) as important for protection from DSS-induced inflammation and control of intestinal *E. coli* levels. They go on to observe that the lectin domain-containing protein Reg4 binds to bacterial LPS and stimulates serum-mediated killing of *E. coli*, but only in the presence of CFD. Lastly, the study shows that mice that lack Reg4 have the same inability to control *E. coli* levels as mice deficient in CFD. The authors conclude that Reg4 binding to *E. coli* mediates CFD-dependent complement-mediated bactericidal activity, and that this mechanism can be used by the host to suppress blooms of Enterobacteriaceae during inflammation.

The potential role of the complement system in controlling luminal intestinal microorganisms is controversial, as there are several conflicting reports regarding the expression and/or presence of complement components in the intestinal mucosa. The current study is thus certainly of interest as it adds to the debate surrounding this issue. Furthermore, identification of a potentially important function for Reg4 is also significant.

While elements of the investigation are convincing, in its current form the manuscript has multiple issues that must be resolved in order to support the author's conclusions.

General points:

1. There is an overall lack of detailed explanation or clear interpretation of the experimental data presented in the main text and figure legends that make the manuscript very hard to follow. For example, Ext. data Fig.4C-E shows ELISAs combined with blots for several proteins from four different mouse genotypes, with four different sample types per genotype, but does not mention the different sample types at all or fully explain what the blots actually show. There are similar issues related to Fig1I-J, Ext. data Fig.1, Fig. 3G-H and Fig. 6. I would suggest that the authors go through their whole results section and ensure that all of the presented experiments are sufficiently rationalised and the results explained clearly.
2. The standard of English in the manuscript is quite variable and could be improved. This includes (but is not limited to) specific errors such as reference to 'membrane attach complexes' instead of 'attack complexes'. The phrase 'switch bactericidal activity on E. coli' in the title does not make sense. Do the authors mean 'direct bactericidal activity towards E. coli' or similar?
3. Loading controls are missing from Western blots shown in Fig.1A, 2H and Ext. data 1C/F.
4. All of the immunofluorescent histology images are poor quality (e.g. small or blurry with low resolution) and must be improved to be convincing.
5. The methods section needs to be significantly clarified, for example by subdividing elements such as the 'Complement assays' section into different types of experiment.

Specific issues:

6. The authors are not clear about where they believe Reg4/CFD mediated bactericidal activity is occurring. Their Fig.2 microbiota sequencing data and Fig.4 E. coli quantification data suggests that Reg4/CFD are acting on the luminal E. coli population, and the authors show detection of luminal CFD in Ext. data Fig.1C. However, in their imaging data (Fig.2G, 6A) they focus on what appears to be tissue invasive E. coli. Where do the authors think this system is active? If the authors believe the activity is luminal then they must show imaging data supporting Reg4/complement binding to luminal E. coli.
7. The authors claim that increased E. coli in CFD and Reg4 KO mice is due to loss of bactericidal activity, supported by the fact that Reg4 and CFD can cooperatively enhance complement-mediated bacterial lysis in vitro. However, it is also quite possible that alterations in the mucosal environment in these mice modifies the growth of E. coli independent of bactericidal activity. The authors should try to show that bactericidal activity is actually altered in vivo, for example by quantifying live/dead E. coli cell ratios in vivo or by using an auxotrophic E. coli model that is incapable of replicating in vivo.
8. RE Fig.1 – The comparison in (A) should be performed by qRT-PCR. Is the CFDKO colon image (B) actually from colon? It looks more like small intestine.
9. RE Ext. data Fig.1 – Presumably the GF mice are WT as well, so the comparison should be between ConvR and GF. In the Western blots in panel (F), what are the high molecular weight bands that appear in some of the samples?
10. RE Fig.2 – The microbiota data shown in (B-D) should be more clearly organized, e.g. charts should be labelled with genotype and treatment. Statistical comparison of microbiota data would also be useful, is it only the E. coli that is changing in the CFD KO mice? The antibody used in (E) is not

detailed in the methods. The method for bacterial DNA extraction is not described.

11. RE Fig.3 – The differences between WT and CFDKO mice in the DSS experiments presented here are much less clear than the data shown in Fig.1, how reproducible is this phenotype? There should be statistical comparisons between the non-colonized and E. coli O160 colonized mice from both genotypes.

12. RE Fig.4 – Are these colonization experiments still performed on animals undergoing DSS treatment as with all the previous experiments? If not, why not? There is no detail in methods regarding the generation of the GFP-E. coli strain, if the GFP label is plasmid based then in vivo plasmid stability quantification must be shown.

13. RE Ext. data Fig.4 – Blots shown in (C-E) require WT control images. Data from inflammasome-related KO tissues is difficult to interpret without littermate control samples.

14. RE Fig.5 – Control experiments with only the recombinant Reg4 constructs must be done to show that the protein has no intrinsic bactericidal activity. More detail regarding the Reg4 mutant must be provided, e.g. where is the protein mutated, does the mutation affect binding, etc.?

15. RE Fig.6 – The use of antibiotic treatments in these experiments is not explained. Images from WT animals should be shown in (A). Binding experiment methods in (C-D) are not clearly detailed.

16. RE Fig.7 – Statistical comparison of microbiota data is needed. Are the Reg4 mice also more susceptible to DSS?

A point by point reply:

Reviewer #1 (Remarks to the Author):

In this manuscript, Qi et al demonstrated that both gut epithelial cell-derived CFD and Reg4 play important roles in eliminating E. coli and thus maintaining gut homeostasis, through mechanisms mainly involving the complement-mediated attach complexes. In general, the topic is novel, interesting and important. However, the results presented in this manuscript are loosely connected with each other, and the mechanisms described are quite broad and multi-dimensional rather than deeply investigated. There are two major issues that the authors need to be concerned about:

What is the relationship between CFD and Reg4? The authors seem to separate them into 2 different stories in the manuscript, although both of which are claimed to be associated with complement-mediated activity.

Reply, Our goal was to determine the role of compliment system in eliminating E. coli in colon. To realize this, we first generated gut CFD (complement factor D, a specific serine protease that cleaves its unique substrate to generate C3 convertases C3(H₂O)Bb and C3bBb) conditional knockout mice (CFD^{fl/fl}pvillin-cre^T mice) and observed the effects of CFD deficiency on E. coli. Then, we investigated how to switch on complement system in colon. We found the REG4, expressed in gut epithelial cells, was involved in initiating the complement-mediated killing on E. coli. Finally, we generated REG4 KO mice to demonstrate the role of Reg4 in compliment system-mediated killing on E coli.

To establish the relationship between CFD and Reg4, we not only offer evidence that Reg4 promotes the killing of E. coli in normal serum but not CFD deficient sera but also give indirect evidences such as the binding of Reg4 with LPS (Fig. 6c), the deposition of REG4 on the *E. coli in vivo* (Fig. 6a), the binding of *E. coli* with Reg4 (Fig. 6b) and that Reg4 could cause the deposition of C3 (Fig. 6d, e).

Taken together, this manuscript should describe a complete story.

The only evidence showing the relationship between CFD and Reg4 is that Reg4/REG4 promotes the killing of E.coli in normal serum but not CFD deficient sera in vitro (Figure 5f), so it is insufficient and not convincing to conclude that “We also found that gut epithelia cells derived CFD play a critical role in Reg4-mediated bactericidal activity on the E. coli” (line 283 and 284).

Reply, Actually, beyond the evidence that Reg4/REG4 promotes the killing of E.coli in normal serum but not CFD deficient sera (Figure 5f), we also made following observations: 1) We used ELISA to examine the binding of Reg4 with LPS (Fig. 6c); 2) We also detected deposition of REG4 on the *E. coli in vivo* (Fig. 6a); 3) Flow cytometry also showed the binding of *E. coli* with Reg4 (Fig. 6b); 4) Reg4 could

cause the deposition of C3 (Fig. 6d, e). Although these are indirect relationship between CFD and Reg4, data indeed suggest that Reg4 is an initiator of complement system.

However, we have already moved out “We also found that gut epithelia cells derived CFD play a critical role in Reg4-mediated bactericidal activity on the *E. coli*”

How about the activity of CFD in Reg4 deficient mice and vice versa?

Reply, We have performed some observations about the activity of CFD in Reg4 deficient mice and vice versa. 1) CFD activity in Reg4 deficient mice was not different from wt mice (Fig. 1 a and b in this point by point reply); 2) There are significantly more Reg4 bound *E. coli* in CFD KO mice, but reg4 expression is not difference between wt and CFD KO (Fig. 1 c and d). More REG4 bound *E. coli* is from CFD deficiency, which could not cause *E. coli* killing.

Fig.1. Activity of CFD in Reg4 deficient mice and vice versa. a. qRT-PCR of CFD in the colon tissues of wt and Reg4 ko mice; b. C3/C3b ratio in colon tissues (left) and in feces (right). C3 and C3b were detected using Elisa, and then ratio of C3 and C3b was calculated. c, qRT-PCR of Reg4 in the colon tissues of wt and CFDKO mice. d, Immunostaining of Reg4 bound *E. coli* in colon tissues. Reg4 bound *E. coli* were compared between wt and CFD KO mice. “**” < 0.05; NS, no significance.

Does the deletion of CFD in Reg4 deficient mice influence the phenotypes and vice versa?

Reply, These are good suggestions. We will do these in the future.

2. The authors tried to explain the DSS and *E. coli* phenotypes in CFDKO vs. WT mice through different mechanisms involving inflammatory macrophages,

inflammasome-related protein production (IL18, NLRC4, caspase1, etc) and complement-mediated bactericidal activity.

Can the authors clarify whether these mechanisms are related with each other and what is the key mechanism in mediating these phenotypes?

Reply, Our data should clearly suggest a mechanism: Since complement-mediated bactericidal activity decreased, this caused increased *E. coli* in CFD deficient mice (Fig. 2). Increased *E. coli* in CFD deficient mice stimulated gut epithelial cell NLRC4 and CASP1 inflammasome to produce IL-18 (Fig. 4). Produced IL-18 by gut epithelial cells could cause inflammatory macrophages through IFN- γ producing cells.

As described in the text, “Since the generation of inflammatory macrophages in gut tissues is dependent on IFN γ (Immunity 2009, **31**(4): 539-550), we also analyzed the IFN γ -producing cells. A drastic expansion of interferon γ (IFN γ)-producing CD4⁺ T helper cells cells (CD4⁺IFN γ ⁺Th1 cells) was observed in *E. coli* infused CFD^{fl/fl}pvillin-cre^Tmice (Fig. 3h). NK cells, another IFN γ producing cells also accumulated in gut tissues of *E. coli* infused CFD^{fl/fl}pvillin-cre^Tmice (Fig. 3h)”.

There are some minor critiques:

*1. In line 108, 156, 160, the authors use “infusion” to describe fecal microbiome transplantation in germ-free mice, do the authors mean “colonize”? Please also provide evidence that germ-free mice are successfully colonized with mouse feces, *E. coli* or lactobacilli.*

Reply, Yes, mean “colonization”. For demonstrating colonization of bacteria, we have performed aerobic culture and anaerobic culture for germ-free mouse feces. Results were shown in following fig. 2.

Fig. 2. Germ-free mice are colonized with E. coli or lactobacilli.

Germ-free (GF) mice were infused by E. coli (GF/E.coli) or lactobacillus (GF/Lac), and then feces were measured for CFU of bacteria after one week. Feces (10 mg) were diluted in the 1 ml PBS, and then 100 µl was put into plate. After aerobic culture for E.coli (a) and anaerobic culture (b, for lactobacillus), CFUs were counted. “***” < 0.05.

2. In Figure 1G, why the histological scores of some mice exceed 8 if calculated by summing up epithelium and infiltrations scores (maximum scores of each is 4 as described in the methods)? Also, the authors stated in the figure legends that there are 3 slides/mouse and n=6 mice per group, but the number of samples presented in figure 1G apparently doesn't fit. Please clarify.

Reply, 1) As described in the materials and methods, total histological score (epithelium (E) + infiltration (I)) should be “< or = 8; Our data did not exceed 8.
2) We have already corrected data in Fig. 1G.

3. Since the DSS colitis model is strongly related with gut microbiome, is the DSS phenotype in CFDKO mice due to differences in the microbiome between KO and WT mice? The authors state that there are no remarkable changes in proportion of Proteobacteria between steady states KO and WT littermates, what about the other main genera? It is important to clarify if there is difference in overall microbiome configuration between KO and WT mice before any treatment, using parameters like alpha diversity, beta diversity and taxonomy composition.

Reply, 1) Yes, as described in Figure 1 and 2 in the manuscript, DSS phenotype in CFD mice is due to difference in the microbiome between KO and WT mice.
2) Actually, there are different in the proportion of proteobacteria between steady KO and WT littermates (Fig. 2b in the manuscript).
3) As shown in supplementary Fig. S3, there indeed exist difference in multiple genera such as lactobacillus.

4. In Figure 4f, g, are the C1q, C3, C5b-9 and FB levels measured in non-treated or treated CFDKO/WT mice? Please clarify.

Reply, In non-treated CFDKO/WT mice.

5. The authors seem to emphasize that CFD and Reg 4 are mainly expressed in Paneth cells. However, the use of “villin-cre” mice can only demonstrate the role of CFD/ Reg 4 in all epithelial cells, not only Paneth cells. This is an acceptable limitation but needs to be mentioned.

Reply, We have already mentioned this “knockout in mouse gut epithelial cells” (line 115-116).

6. There are some grammar mistakes in the manuscript. E.g, line 115 should be “was selectively knocked out”; line 136 should be “proportion”; line 217 should be “sensitive”; line 242, should be “MASP” instead of “WASP”. Please check thoroughly and correct the grammar mistakes.

Reply, We have already corrected these.

Reviewer #2 (Remarks to the Author):

In the manuscript “Gut epithelial cell derived Reg4 and complement factor D switch bactericidal activity on E. coli to maintain gut homeostasis” by Qi et al, the authors describe a protective role for complement factor D expression by intestinal epithelial cells during intestinal inflammation, in part by limiting growth of enteric E. coli. Intestinal epithelial-specific complement factor D conditional knockout mice (CFD^{fl/fl}-villin-cre, herein referred to as CFDKO mice as described in the manuscript) were generated and exposed to the DSS model of experimental colitis, which resulted in increased mortality and morbidity associated with elevated colonic inflammatory cytokines and lamina propria macrophages. In addition, DSS-treated CFDKO mice displayed alterations in intestinal microbiota, including increased Proteobacteria and Escherichia-Shigella. Transfer of E. coli exacerbates DSS-mediated colitis and in both SPF and germ-free CFDKO mice, accompanied by an increase in E. coli growth in CFDKO mice, suggesting epithelial CFD promotes bactericidal activity in vivo. CFDKO mice display reduced C3b expression and binding to E. coli, and serum from CFDKO mice displayed diminished capacity to restrict E. coli growth in vitro. Interestingly, Reg4 enhanced the bactericidal activity of wild type serum, but failed to do so in CFDKO serum, suggesting this c-type lectin requires CFD for killing of E. coli. Lastly, the authors show that Reg4 KO mice display a similar increased susceptibility to DSS colitis, increased E. coli expansion and reduced C3b deposition on E. coli, indicating that Reg4 functions with CFD to restrict E. coli via complement-mediated killing. The authors provide compelling evidence for a protective role for intestinal epithelial-derived CFD during intestinal inflammation and limiting overgrowth of E. coli, however additional controls and experiments will help shed mechanistic insight on the cellular contributors as well as the order of events by which these events proceed.

Major points

1. In figure 1, the authors convincingly show that CFDKO mice display increased intestinal inflammation during DSS colitis. Interestingly, untreated CFDKO mice also exhibit increased intestinal macrophages compared to naïve wild type littermates. Does this also result in elevated colonic TNF, IL6 and IL1b mRNA, similar to

DSS-treated mice?

Reply, Yes, there have moderately elevated TNF α , IL-6 and IL-1 β as shown in following fig.3. However, we did not observe remarkable changes in the phenotypes of normal CFD KO mice as compared to wt mice (Fig. 5 in this point by point reply)

Fig. 3. qRT-PCR of TNF α , IL-6 and IL-1 β in the colon tissues of normal Wt and CFD KO mice. Wt, wild type mice; CFD KO, CFD KO mice. “*”, p < 0.05; R.E, relative expression.

2. Similar to above, the authors show DSS-treated CFDKO mice have increased Proteobacteria and *E. coli* compared to DSS-treated wild type mice. The untreated CFDKO mice have modest (non-significant?) difference in Proteobacteria, but it is unclear from Extended Fig. 3a if they also have elevated *E. coli* prior to DSS. Were these levels detectable? This control is important for understanding if the loss of CFD limits the overgrowth of *E. coli* in naïve animals, which may contribute to the increased lamina propria macrophages during both homeostasis and intestinal inflammation. An elevation in baseline *E. coli* in CFDKO mice would also be consistent with the comparable levels of pro-inflammatory cytokines and macrophages between *E.coli*-fed wild type mice and non-colonized CFDKO mice (Fig. 3f-g).

Reply, The untreated CFDKO mice have modest difference in Proteobacteria (Fig. 2 b); but it indeed is undetectable for *E.coli* prior to DSS.

3. In Extended Fig. 4c, the authors compare IL-18 expression in *E. coli*-fed *Nlr4* and caspase-1/11 knockout mice to CFDKO mice and conclude that the increase in IL-18 observed in CFDKO mice is due to *Nlr4* and caspase-1/11. However, this cannot be concluded unless the authors target *Nlr4* and caspase-1/11 in the CFDKO mice, similar to the caspase-8 and PKCd inhibitor studies shown in Extended figure 4d. This would require making double knockout animals of CFD and *Nlr4* or caspase-1/11 to conclude this. However, a comparison of *Nlr4* and caspase-1/11 to wild type animals would allow the authors to determine if their strain of *E. coli* activates IL-18 via

Nlrc4 and caspase-1/11. This could implicate the involvement of these proteins during E. coli recognition, but would not fully address if the increased IL-18 in the CFDKO mice is through Nlrc4 and caspase-1/11.

Reply, We totally agree with these. Indeed, we did not fully address the increased IL-18 in the CFD KO mice was through NLRC4 and caspase-1/11 in gut epithelial cells.

4. The authors conclude that CFD regulates E. coli killing via membrane attack complex (MAC) formation. They nicely measure deposition of the early and central components that initiate the complement cascade (i.e., C3b and MASPs), but do not measure differences in deposition of the downstream C5b-C9 components on E. coli, which compose the MAC.

Reply, We also detected downstream C5b-C9 components on E. coli. Data have been added into Fig. 6a in the manuscript.

5. Similar to above, a major function of C3b is to act as an opsonin for phagocytic cells. Given the authors demonstrate increases in intestinal macrophages in CFDKO mice, it stands to reason that CFD may also regulate E. coli-associated intestinal inflammation by promoting C3b-mediated phagocytosis and killing by macrophages. The authors should explore this in vitro by performing macrophage killing assays with E. coli pretreated with serum from wild type vs. CFDKO mice. This experiment would help to connect the author's striking macrophage results in vivo with their proposed mechanism of CFD.

Reply, We try to observe the killing of macrophages on E. coli, wt sera-pretreated E. coli and CFDKO sera-pretreated E. coli. However, it is hard to distinguish the role of macrophages from complements in sera when E. coli are pretreated using sera.

Minor points

In Figure 6a, the authors show microscopy results using CFDKO, but not wild type controls. Please include this result for improved comparisons. In addition, figure 6a presents an interesting finding that CFDKO mice have increased presence of Reg4-associated E. coli. Is this due to an overall increase in Reg4 in the CFDKO mice? This could be addressed by the above request for wild type image comparisons.

Reply, 1) We have already added wild type control;
2) Increased presence of Reg4-associated E coli in the CFD KO mice is from CFD deficiency.

As E. coli can induce intestinal inflammation on its own, do the inflammation-prone

CFDKO mice display greater symptoms of disease/pathology when colonized with the 0160 strain of E. coli (in the absence of DSS)?

Reply, We did not find remarkable phenotype changes such as body weight and disease index in wt and CFD when colonized with the 0160 strain of *E. coli* in the absence of DSS (Fig. 4 in this point by point reply).

Fig. 4. Survival curve and disease activity index (DAI) in wt or CFD KO mice colonized by 0160 E.coli. Wt (WT0160) and CFD KO (CfDKO 0160) mice were infused by 5×10^6 *E. coli* with demonstrated colonization, and then survival rate and disease activity index were observed.

In Figure 7e, the y axis of the flow plots display “Reg4”, which would indicate that some of the presented Reg4KO mice only slightly lower levels of Reg4-bound E. coli compared to wild type. Is this label correct as one would expect Reg4 to be absent in these mice?

Reply, Label is right. Results also exhibited that REG4 bound *E. coli* did not appear in Reg4KO mice.

Reviewer #3 (Remarks to the Author):

In the current manuscript, Qi et al. investigate how the intestinal mucosa might control the blooms of Enterobacteriaceae (e.g. E. coli) that have been described to occur in several instances of inflammation-associated dysbiosis. They investigate a potential role for complement-mediated bactericidal activity in this process and identify epithelial production of complement factor D (CFD) as important for protection from DSS-induced inflammation and control of intestinal E. coli levels. They go on to observe that the lectin domain-containing protein Reg4 binds to bacterial LPS and stimulates serum-mediated killing of E. coli, but only in the presence of CFD. Lastly, the study shows that mice that lack Reg4 have the same inability to control E. coli levels as mice deficient in CFD. The authors conclude that Reg4 binding to E. coli mediates CFD-dependent complement-mediated bactericidal activity, and that this mechanism can be used by the host to suppress blooms of Enterobacteriaceae during inflammation.

The potential role of the complement system in controlling luminal intestinal microorganisms is controversial, as there are several conflicting reports regarding the expression and/or presence of complement components in the intestinal mucosa. The current study is thus certainly of interest as it adds to the debate surrounding this issue. Furthermore, identification of a potentially important function for Reg4 is also significant.

While elements of the investigation are convincing, in its current form the manuscript has multiple issues that must be resolved in order to support the author's conclusions.

General points:

1. There is an overall lack of detailed explanation or clear interpretation of the experimental data presented in the main text and figure legends that make the manuscript very hard to follow. For example, Ext. data Fig.4C-E shows ELISAs combined with blots for several proteins from four different mouse genotypes, with four different sample types per genotype, but does not mention the different sample types at all or fully explain what the blots actually show. There are similar issues related to Fig11-J, Ext. data Fig.1, Fig. 3G-H and Fig. 6. I would suggest that the authors go through their whole results section and ensure that all of the presented experiments are sufficiently rationalised and the results explained clearly.

Reply, We have already improved them.

2. The standard of English in the manuscript is quite variable and could be improved. This includes (but is not limited to) specific errors such as reference to 'membrane attach complexes' instead of 'attack complexes'. The phrase 'switch bactericidal activity on E. coli' in the title does not make sense. Do the authors mean 'direct bactericidal activity towards E. coli' or similar?

Reply, We have already improved these.

3. Loading controls are missing from Western blots shown in Fig.1A, 2H and Ext. data 1C/F.

Reply, Since samples are from feces, it is hard to add loading control such as actin for western blots. However, we put isotypic antibody control.

4. All of the immunofluorescent histology images are poor quality (e.g. small or blurry with low resolution) and must be improved to be convincing.

Reply, We have already improved these.

5. The methods section needs to be significantly clarified, for example by subdividing elements such as the ‘Complement assays’ section into different types of experiment.

Reply, We have already improved this.

Specific issues:

6. The authors are not clear about where they believe Reg4/CFD mediated bactericidal activity is occurring. Their Fig.2 microbiota sequencing data and Fig.4 E. coli quantification data suggests that Reg4/CFD are acting on the luminal E. coli population, and the authors show detection of luminal CFD in Ext. data Fig.1C. However, in their imaging data (Fig.2G, 6A) they focus on what appears to be tissue invasive E. coli. Where do the authors think this system is active? If the authors believe the activity is luminal then they must show imaging data supporting Reg4/complement binding to luminal E. coli.

Reply, 1) The activity of CFD is in the luminal. But, E.coli were analyzed under two different situations, normal and 2% DSS treated mice. In normal mice, we detected luminal E.coli population in feces; In 2% DSS treated mice, we detect E.coli in colon tissues such as Fig2G, Fig.6A.

2) For luminal E. coli, we have already shown imaging data supporting Reg4/complement binding to luminal E. coli (Figure 6a, b and Figure 7e)

7. The authors claim that increased E. coli in CFD and Reg4 KO mice is due to loss of bactericidal activity, supported by the fact that Reg4 and CFD can cooperatively enhance complement-mediated bacterial lysis in vitro. However, it is also quite possible that alterations in the mucosal environment in these mice modifies the growth of E. coli independent of bactericidal activity. The authors should try to show that bactericidal activity is actually altered in vivo, for example by quantifying live/dead E. coli cell ratios in vivo or by using an auxotrophic E. coli model that incapable of replicating in vivo.

Reply, 1) Indeed, it is also possible that alterations in the mucosal environment in these mice modify the growth of E. coli independent of bactericidal activity. To avoid this, we generated germ-free CFD KO and WT mice (Fig. 4). This should eliminate the alteration in these mice may modify the growth of *E.coli*.

2) It is hard to quantify live/dead *E.coli* cell ration in vivo. Although live/dead bacterial viability kits are available, it is only used in vitro for short time.

3) We will generate an auxotrophic E. coli model that is incapable of replicating in vivo in the future.

8. *RE Fig.1 – The comparison in (A) should be performed by qRT-PCR. Is the CFDKO colon image (B) actually from colon? It looks more like small intestine.*

Reply, 1) We performed qRT-PCR, and results was put in the Fig. 1A.
2) The CFDKO colon images indeed came from colon.

9. *RE Ext. data Fig.1 – Presumably the GF mice are WT as well, so the comparison should be between ConvR and GF. In the Western blots in panel (F), what are the high molecular weight bands that appear in some of the samples?*

Reply, 1) Indeed, GF mice are from conventional wt mice.
2) The modified figures were put into supplementary Fig. 1.

10. *RE Fig.2 – The microbiota data shown in (B-D) should be more clearly organized, e.g. charts should be labelled with genotype and treatment. Statistical comparison of microbiota data would also be useful, is it only the E. coli that is changing in the CFD KO mice? The antibody used in (E) is not detailed in the methods. The method for bacterial DNA extraction is not described.*

Reply, 1) We have already further organized the data in Fig. 2.
2) Statistical analyses have been done in Fig. 2.
3) FITC labeled anti- LPS antibody was described in supplementary Table S1.
4) The methods for bacteria DNA extraction have been added into method (gut microbiota analyses).

11. *RE Fig.3 – The differences between WT and CFDKO mice in the DSS experiments presented here at much less clear than the data shown in Fig.1, how reproducible is this phenotype? There should be statistical comparisons between the non-colonized and E. coli O160 colonized mice from both genotypes.*

Reply, 1) As described in the manuscript, different concentrations of DSS was used. In figure 1, 2.5 % DSS was used; whereas 2% DSS was used in Figure 3 (Since colonization of E.coli potentially increase sensitive to DSS-mediated colitis, 2% DSS was used in this experiment).
2) We have already performed statistical comparison between the non-colonized and E. coli 0160 colonized mice between both genotypes (Fig. 3 in the manuscript)

12. *RE Fig.4 – Are these colonization experiments still performed on animals undergoing DSS treatment as with all the previous experiments? If not, why not? There is no detail in methods regarding the generation of the GFP-E. coli strain, if the GFP label is plasmid based then in vivo plasmid stability quantification must be shown.*

Reply, 1) No DSS-treatment. The goal of this experiment is to observe the act of CFD and REG4 on the luminal bacteria, thus it was not necessary for DSS-treatment.
2) We added the method for the generation of the GFP-E into Fig. 4 legend.

13. RE Ext. data Fig.4 – Blots shown in (C-E) require WT control images. Data from inflammasome-related KO tissues is difficult to interpret without littermate control samples.

Reply, We modified this figure and added control also (Complementary Fig. 4).

14. RE Fig.5 – Control experiments with only the recombinant Reg4 constructs must be done to show that the protein has no intrinsic bactericidal activity. More detail regarding the Reg4 mutant must be provided, e.g. where is the protein mutated, does the mutation affect binding, etc.?

Reply, 1) Control experiments have already been done, relative data were added into Figure 5d.
2) Mutant hREGIV-P91S was described in methods (Generation of mouse Reg4 and human REG4 and mutant human REG4).
3) The reduced binding by mutated protein was shown (Fig. 5 in this point by point reply).

Fig. 5. Flow cytometry of REG4 and mutant REG4 stained cells. GFP-labeled E. coli were stained by recombinant REG4 (1 μ g) or mutated REG4 (1 μ g), and then secondary antibodies. “***” <0.01.

15. RE Fig.6 – The use of antibiotic treatments in these experiments is not explained. Images from WT animals should be shown in (A). Binding experiment methods in (C-D) are not clearly detailed.

Reply, 1) The explain for use of antibiotic treatments have been added (line 379-380);
2) Images from WT animals have been shown in the Fig. 6 A in the manuscript;
3) Binding experiment methods have been further described in methods (Complement assays).

16. RE Fig.7 – Statistical comparison of microbiota data is needed. Are the Reg4 mice also more susceptible to DSS?

Reply, 1) We performed statistical comparison of microbiota data in the Fig. 7 in the manuscript;
2) Reg4 mice are more susceptible to DSS mediated colitis (Fig. 6 in this point by point reply). These data appear in another paper.

Fig. 6. REG4 deficient mice are more sensitive to DSS-mediated colitis.

a, b and **c** The survival rate (a), body weight (b) and disease index (c) in REG4 KO mice and their littermate WT mice (male, n=15) after DSS (2.5%) treatment.
d The length of colon in REG4KO and their littermate WT after DSS (2.5%) treatment. **e** H&E staining and histology score of colon tissues of REG4KO and their littermate WT after DSS (2.5%) treatment. For histological score, 3 slides/mouse, n=6. Scale bar, 40 μ M.

Reviewers' comments:

Reviewer #1 (Remarks to the Author):

In general, Qi et al have addressed my previous major and minor concerns and make the manuscript more logically connected.

I have two minor suggestions regarding the revision:

1. It will be better to include figure 1 in point-to-point response (Activity of CFD in Reg4 deficient mice and vice versa) in supplemental material and briefly addressed this in the manuscript. Furthermore, the y axis label in Figure 1b should be "C3/C3b ratio" rather than "C3/C3b ration".
2. Please include figure 1 in point-to-point response (Germ-free mice are colonized with E. coli or lactobacilli) to show that the colonization in the germ-free mice is successful.

Reviewer #2 (Remarks to the Author):

In the revised version of the manuscript, the authors do not provide data using CFDKO mice crossed with either Nlrc4 or caspase-11 knockout mice to support the contribution of Nlrc4/caspase-11 during CFD-mediated control of IL-18. Instead they tone down the language describing this relationship, which is a satisfactory revision.

They do provide the requested and necessary experimental control involving E.coli-fed WT mice to help implicate the requirement of these proteins during IL-18 production, however no differences are observable between the E. coli-treated wild type and CFDKO mice (Supplementary figure 4c-e). These results do not support the author's conclusion that "Higher levels of mature IL-18 were present in the colonic tissues in CFDfl/flpvillin-creTmice after giving E. coli stimulation (Pgs 7+8, line 171-172). Moreover, these experiments would argue that CFD is not involved in the regulation of IL-18 following E. coli exposure, and likely other additional factors are necessary for driving the exaggerated IL-18 production in CFDKO mice during DSS colitis. This important points needs to be amended and addressed in the revised manuscript.

In point 5, the authors were also asked to explore the contribution of complement opsonization of E. coli during macrophage-mediated killing in order to provide better mechanistic links between their primary observations. However, the conclusion in their response "it is hard to distinguish the role of macrophages from complements in sera when E. coli are pretreated using sera" is somewhat confusing. Were the authors unable to observe measurable differences in macrophage killing between WT vs. CFD serum-treated E. coli or where there technical limitations preventing the completion of these experiments? Providing the data or additional details would help clarify this point.

Minor comment:

On page 7, lines 157-160, the author's state "After the equal amount of E. coli 0160 were infused into mice, CFDfl/fl pvillin-creT mice had also more severe colitis than CFDfl/flpvillin-creW mice after feeding 2 % DSS, including higher survival rate, more weight loss, higher disease index and more shortening colon (Fig. 3a-d)." This should be "lower survival rate"

Reviewer #3 (Remarks to the Author):

The authors have provided a point-by-point reply to the issues that were raised in the first round of revision. While some of these issues have been examined in the new submission, there are still multiple points that must be addressed to support the claims that the authors make in this paper. Below are point-by-point responses to the author's replies to the first review. All new reviewer responses are prefaced by 'Reviewer response'.

General points:

1. There is an overall lack of detailed explanation or clear interpretation of the experimental data presented in the main text and figure legends that make the manuscript very hard to follow. For example, Ext. data Fig.4C-E shows ELISAs combined with blots for several proteins from four different mouse genotypes, with four different sample types per genotype, but does not mention the different sample types at all or fully explain what the blots actually show. There are similar issues related to Fig1I-J, Ext. data Fig.1, Fig. 3G-H and Fig. 6. I would suggest that the authors go through their whole results section and ensure that all of the presented experiments are sufficiently rationalised and the results explained clearly.

Author Reply: We have already improved them.

Reviewer response: This is partly improved but still needs more work. The data presented in Supplementary Fig.1 and in Fig. 6 should be described in more detail.

2. The standard of English in the manuscript is quite variable and could be improved. This includes (but is not limited to) specific errors such as reference to 'membrane attach complexes' instead of 'attack complexes'. The phrase 'switch bactericidal activity on E. coli' in the title does not make sense. Do the authors mean 'direct bactericidal activity towards E. coli' or similar?

Author Reply, We have already improved these.

Reviewer response: Also partly improved, but the incorrect phrase 'membrane attach complex(s)' still occurs multiple times throughout the manuscript, please change this.

3. Loading controls are missing from Western blots shown in Fig.1A, 2H and Ext. data 1C/F.

Author Reply. Since samples are from feces, it is hard to add loading control such as actin for western blots. However, we put isotypic antibody control.

Reviewer response: Isotype antibody controls are not the same thing as loading controls. I understand that specific blotting for standard loading control proteins is not appropriate in fecal preparations, but the authors could stain WB membranes for total protein (e.g. Coomassie, Ponceau etc.) to validate equal protein loading between samples.

4. All of the immunofluorescent histology images are poor quality (e.g. small or blurry with low resolution) and must be improved to be convincing.

Author Reply, We have already improved these.

Reviewer response: It is not immediately clear what has been improved. The removal of high

magnification images from Fig.6a has made this data less convincing.

5. The methods section needs to be significantly clarified, for example by subdividing elements such as the 'Complement assays' section into different types of experiment.

Author Reply, We have already improved this.

Reviewer response: A little more detail is now provided, but for clarity it would be useful to separate the different complement-related experiments into separate method sections, as suggested in the original review comment.

Specific issues:

6. The authors are not clear about where they believe Reg4/CFD mediated bactericidal activity is occurring. Their Fig.2 microbiota sequencing data and Fig.4 E. coli quantification data suggests that Reg4/CFD are acting on the luminal E. coli population, and the authors show detection of luminal CFD in Ext. data Fig.1C. However, in their imaging data (Fig.2G, 6A) they focus on what appears to be tissue invasive E. coli. Where do the authors think this system is active? If the authors believe the activity is luminal then they must show imaging data supporting Reg4/complement binding to luminal E. coli.

Author Reply, 1) The activity of CFD is in the luminal. But, E.coli were analyzed under two different situations, normal and 2% DSS treated mice. In normal mice, we detected luminal E.coli population in feces; In 2% DSS treated mice, we detect E.coli in colon tissues such as Fig2G, Fig.6A.

2) For luminal E. coli, we have already shown imaging data supporting Reg4/complement binding to luminal E. coli (Figure 6a, b and Figure 7e)

Reviewer response: The authors need to be very clear at different points in the manuscript text and in the figure legends when they are analysing luminal and tissue associated E. coli, as it is possible that factors deposited on tissue invasive bacteria are not derived from the intestinal epithelium. I understand that the flow cytometry data in Fig. 6b and 7e indicate that these factors can be detected on luminal E. coli, but imaging data would still be useful to support this. I am unable to see any luminal E. coli in Fig. 6a, which the authors reference in their response.

7. The authors claim that increased E. coli in CFD and Reg4 KO mice is due to loss of bactericidal activity, supported by the fact that Reg4 and CFD can cooperatively enhance complement-mediated bacterial lysis in vitro. However, it is also quite possible that alterations in the mucosal environment in these mice modifies the growth of E. coli independent of bactericidal activity. The authors should try to show that bactericidal activity is actually altered in vivo, for example by quantifying live/dead E. coli cell ratios in vivo or by using an auxotrophic E. coli model that incapable of replicating in vivo.

Author Reply, 1) Indeed, it is also possible that alterations in the mucosal environment in these mice modify the growth of E. coli independent of bactericidal activity. To avoid this, we generated germ-free CFD KO and WT mice (Fig. 4). This should eliminate the alteration in these mice may modify the growth of E.coli.

2) It is hard to quantify live/dead E.coli cell ration in vivo. Although live/dead bacterial viability kits are available, it is only used in vitro for short time.

3) We will generate an auxotrophic E. coli model that is incapable of replicating in vivo in the future.

Reviewer response: The authors are correct that using GF mice removes the possibility that alterations in the microbiota might influence the mucosal environment; however, this does not correct for alterations in host-derived factors that might occur after loss of CFD or Reg4. Live/dead staining can

be performed directly on bacteria isolated from mouse faeces and be analysed by microscopy or flow cytometry. This would provide clear evidence of alterations in bactericidal activity in vivo.

8. RE Fig.1 – The comparison in (A) should be performed by qRT-PCR. Is the CFDKO colon image (B) actually from colon? It looks more like small intestine.

Author Reply, 1) We performed qRT-PCR, and results was put in the Fig. 1A.
2) The CFDKO colon images indeed came from colon.

Reviewer response: The data in Fig. 1a looks fine; however, the CFDKO colon image is still quite confusing as it does look like small intestine in terms of crypt morphology and what look like villus structures. Can the authors please provide an alternative image that is less ambiguous?

9. RE Ext. data Fig.1 – Presumably the GF mice are WT as well, so the comparison should be between ConvR and GF. In the Western blots in panel (F), what are the high molecular weight bands that appear in some of the samples?

Author Reply, 1) Indeed, GF mice are from conventional wt mice.
2) The modified figures were put into supplementary Fig. 1.

Reviewer response: As both GF and conventionally raised (ConvR) mice are genetically WT, please re-label graphs and figures with ConvR instead of WT. Regarding the high molecular weight bands in the Western blots in this figure, the intention was not for the authors to crop/remove this data but to identify and discuss it. Even if the authors do not determine the nature of these bands, their variable presence should be noted.

10. RE Fig.2 – The microbiota data shown in (B-D) should be more clearly organized, e.g. charts should be labelled with genotype and treatment. Statistical comparison of microbiota data would also be useful, is it only the E. coli that is changing in the CFDKO mice? The antibody used in (E) is not detailed in the methods. The method for bacterial DNA extraction is not described.

Author Reply, 1) We have already further organized the data in Fig. 2.
2) Statistical analyses have been done in Fig. 2.
3) FITC labeled anti- LPS antibody was described in supplementary Table S1.
4) The methods for bacteria DNA extraction have been added into method (gut microbiota analyses).

Reviewer response: Bacterial DNA extraction methods have been expanded, but are still very unclear. For example, which 'bacterial DNA extraction kit' was used? Details for qPCR-based bacterial quantification (instrument, PCR conditions, PCR reagents etc.) are also absent.

11. RE Fig.3 – The differences between WT and CFDKO mice in the DSS experiments presented here at much less clear than the data shown in Fig.1, how reproducible is this phenotype? There should be statistical comparisons between the non-colonized and E. coli O160 colonized mice from both genotypes.

Author Reply, 1) As described in the manuscript, different concentrations of DSS was used. In figure 1, 2.5 % DSS was used; whereas 2% DSS was used in Figure 3 (Since colonization of E.coli potentially increase sensitive to DSS-mediated colitis, 2% DSS was used in this experiment).
2) We have already performed statistical comparison between the non-colonized and E. coli O160 colonized mice between both genotypes (Fig. 3 in the manuscript)

Reviewer response: The rationale for using different concentrations of DSS in these experiments must be explicitly stated in the manuscript. I understand that the authors have presented statistical comparison between WT and CFDKO mice in Fig3a, d-g, but the group comparisons in Fig3b-c are not indicated. More importantly, the original request was for statistical comparison between non-colonized and E. coli O160-colonized mice from each genotype (e.g. WT vs WT/O160 and CFDKO vs CFDKO/O160). This is relevant as it is important to independently discriminate between the effects of genotype and E. coli colonization on DSS sensitivity.

12. RE Fig.4 – Are these colonization experiments still performed on animals undergoing DSS treatment as with all the previous experiments? If not, why not? There is no detail in methods regarding the generation of the GFP-E. coli strain, if the GFP label is plasmid based then in vivo plasmid stability quantification must be shown.

Author Reply, 1) No DSS-treatment. The goal of this experiment is to observe the act of CFD and REG4 on the luminal bacteria, thus it was not necessary for DSS-treatment.
2) We added the method for the generation of the GFP-E into Fig. 4 legend.

Reviewer response: The rationale for not using DSS in this experiment must be explicitly stated in the text. Methods for GFP-E. coli generation should be described in detail in the methods section, not the figure legend. In vivo plasmid stability is quite variable, as bacteria tend to lose plasmids in the absence of selective pressure in vivo (e.g. during mouse colonization), do the authors have any data demonstrating plasmid stability during colonization?

13. RE Ext. data Fig.4 – Blots shown in (C-E) require WT control images. Data from inflammasome-related KO tissues is difficult to interpret without littermate control samples.

Author Reply, We modified this figure and added control also (Complementary Fig. 4).

Reviewer response: The figure is improved, but there are still some issues. In the results text (line 168) Supplementary Fig.4a is referred to as qRT-PCR, but appears to be ELISA based on the legend and presentation in the figure, can the authors clarify this? Supplementary Fig.4c-e seem to show similar levels of IL-18 in vehicle treated WT and CFDKO tissues, but isn't this contradicted by Supplementary Fig.4a? Lastly, can the authors clarify if this data is from in vivo samples or tissues treated ex vivo, as in the original submission?

14. RE Fig.5 – Control experiments with only the recombinant Reg4 constructs must be done to show that the protein has no intrinsic bactericidal activity. More detail regarding the Reg4 mutant must be provided, e.g. where is the protein mutated, does the mutation affect binding, etc.?

Author Reply, 1) Control experiments have already been done, relative data were added into Figure 5d.
2) Mutant hREGIV-P91S was described in methods (Generation of mouse Reg4 and human REG4 and mutant human REG4).
3) The reduced binding by mutated protein was shown (Fig. 5 in this point by point reply).

Reviewer response: Control experiments are convincing. Data shown in Fig.5 of the author reply does show a difference in WT and mutant Reg4 E. coli binding and this should be included in the manuscript. Could the authors comment on why Reg4 seems to only bind a small fraction of the E. coli cells in this data?

15. RE Fig.6 – The use of antibiotic treatments in these experiments is not explained. Images from WT

animals should be shown in (A). Binding experiment methods in (C-D) are not clearly detailed.

Author Reply, 1) The explain for use of antibiotic treatments have been added (line 379-380);
2) Images from WT animals have been shown in the Fig. 6 A in the manuscript;
3) Binding experiment methods have been further described in methods (Complement assays).

Reviewer response: The functional purpose of antibiotic use is obvious, but their use in this context needs to be explicitly stated in the manuscript text. Why is microbiota suppression required in this case? The inclusion of WT images in Fig.6 is useful, but the removal of high magnification images that were present in the original submission makes it impossible for the reader to evaluate the colocalisation of E. coli cells and the different proteins examined. High magnification images should be included for all panels.

16. RE Fig.7 – Statistical comparison of microbiota data is needed. Are the Reg4 mice also more susceptible to DSS?

Author Reply, 1) We performed statistical comparison of microbiota data in the Fig. 7 in the manuscript;
2) Reg4 mice are more susceptible to DSS mediated colitis (Fig. 6 in this point by point reply). These data appear in another paper.

Microbiota comparison and Reg4 KO DSS susceptibility data is convincing.

A point by point reply,

Reviewers' comments:

Reviewer #1 (Remarks to the Author):

In general, Qi et al have addressed my previous major and minor concerns and make the manuscript more logically connected.

I have two minor suggestions regarding the revision:

1. It will be better to include figure 1 in point-to-point response (Activity of CFD in Reg4 deficient mice and vice versa) in supplemental material and briefly addressed this in the manuscript. Furthermore, the y axis label in Figure 1b should be “C3/C3b ratio” rather than “C3/C3b ration”.

Reply, We have put this Figure into supplemental material as Figure S8 and also addressed in the manuscript. Meanwhile, we also corrected y axis label.

2. Please include figure 1 in point-to-point response (Germ-free mice are colonized with E. coli or lactobacilli) to show that the colonization in the germ-free mice is successful.

Reply, We have put Figure into manuscript as Figure 4f.

Reviewer #2 (Remarks to the Author):

In the revised version of the manuscript, the authors do not provide data using CFDKO mice crossed with either Nlrc4 or caspase-11 knockout mice to support the contribution of Nlrc4/caspase-11 during CFD-mediated control of IL-18. Instead they tone down the language describing this relationship, which is a satisfactory revision. They do provide the requested and necessary experimental control involving E.coli-fed WT mice to help implicate the requirement of these proteins during IL-18 production, however no differences are observable between the E. coli-treated wild type and CFDKO mice (Supplementary figure 4c-e). These results do not support the author's conclusion that “Higher levels of mature IL-18 were present in the colonic tissues in CFDfl/flpvillin-creTmice after giving E. coli stimulation (Pgs 7+8, line 171-172). Moreover, these experiments would argue that CFD is not involved in the regulation of IL-18 following E. coli exposure, and likely other additional factors are necessary for driving the exaggerated IL-18 production in CFDKO mice during DSS colitis. This important points needs to be amended and addressed in the revised manuscript.

Reply, We used confusing samples in figure S4. Now, correct results have been added into figure S4.

In point 5, the authors were also asked to explore the contribution of complement opsonization of E. coli during macrophage-mediated killing in order to provide better mechanistic links between their primary observations. However, the conclusion in their response “it is hard to distinguish the role of macrophages from complements in sera when E. coli are pretreated using sera” is somewhat confusing. Were the authors unable to observe measurable differences in macrophage killing between WT vs. CFD serum-treated E. coli or where there technical limitations preventing the completion of these experiments? Providing the data or additional details would help clarify this point.

Reply, We try to observe the killing of macrophages on E. coli, wt sera-pretreated E. coli and CFDKO sera-pretreated E. coli. However, it is hard to distinguish the role of macrophages from the complements in sera. When E. coli are pretreated using sera, component components in sera also kill E. coli; However, results have been shown in Figure 1 in this point by point reply.

Figure 1. Killing of macrophages on wild type (WT) sera treated or CFD KO sera treated E. coli. The E. coli were first pretreated using wt or CFD sera, and then added into macrophage cultures. The supernatants were taken out at different time points, and bacterium numbers were detected by culturing in plates. E. coli, untreated sera.

Minor comment:

On page 7, lines 157-160, the author's state "After the equal amount of E. coli 0160 were infused into mice, CFDfl/fl pvillin-creT mice had also more severe colitis than CFDfl/flpvillin-creW mice after feeding 2 % DSS, including higher survival rate, more weight loss, higher disease index and more shortening colon (Fig. 3a-d)." This should be "lower survival rate"

Reply, We have already corrected this.

Reviewer #3 (Remarks to the Author):

The authors have provided a point-by-point reply to the issues that were raised in the first round of revision. While some of these issues have been examined in the new submission, there are still multiple points that must be addressed to support the claims that the authors make in this paper. Below are point-by-point responses to the author's replies to the first review. All new reviewer responses are prefaced by 'Reviewer response'.

General points:

1. There is an overall lack of detailed explanation or clear interpretation of the experimental data presented in the main text and figure legends that make the manuscript very hard to follow. For example, Ext. data Fig.4C-E shows ELISAs combined with blots for several proteins from four different mouse genotypes, with four different sample types per genotype, but does not mention the different sample types at all or fully explain what the blots actually show. There are similar issues related to Fig1I-J, Ext. data Fig.1, Fig. 3G-H and Fig. 6. I would suggest that the authors go through their whole results section and ensure that all of the presented experiments are sufficiently rationalised and the results explained clearly.

Author Reply: We have already improved them.

Reviewer response: This is partly improved but still needs more work. The data presented in Supplementary Fig.1 and in Fig. 6 should be described in more detail.

Reply, We have further improved this. We have also described the data presented in supplementary Fig.1 and in Fig. 6 in more detail.

2. The standard of English in the manuscript is quite variable and could be improved. This includes (but is not limited to) specific errors such as reference to 'membrane attach complexes' instead of 'attack complexes'. The phrase 'switch bactericidal activity on E. coli' in the title does not make sense. Do the authors mean 'direct bactericidal activity towards E. coli' or similar?

Author Reply, We have already improved these.

Reviewer response: Also partly improved, but the incorrect phrase 'membrane attach complex(s)' still occurs multiple times throughout the manuscript, please change this.

Reply, We have corrected this.

3. Loading controls are missing from Western blots show in Fig.1A, 2H and Ext. data 1C/E.

Author Reply, Since samples are from feces, it is hard to add loading control such as actin for western blots. However, we put isotopic antibody control.

Reviewer response: Isotype antibody controls are not the same thing as loading controls. I understand that specific blotting for standard loading control proteins is not appropriate in fecal preparations, but the authors could stain WB membranes for total protein (e.g. Coomassie, Ponceau etc.) to validate equal protein loading between samples.

Reply, We have already added these loading control.

4. All of the immunofluorescent histology images are poor quality (e.g. small or blurry with low resolution) and must be improved to be convincing.

Author Reply, We have already improved these.

Reviewer response: It is not immediately clear what has been improved. The removal of high magnification images from Fig.6a has made this data less convincing.

Reply, We have further improved the immunofluorescence history image, and high magnification images have also been added into Fig. 6a.

5. The methods section needs to be significantly clarified, for example by subdividing elements such as the 'Complement assays' section into different types of experiment.

Author Reply, We have already improved this.

Reviewer response: A little more detail is now provided, but for clarity it would be useful to separate the different complement-related experiments into separate method sections, as suggested in the original review comment.

Reply, we have already separated the different complement-related experiments into different method section.

Specific issues:

6. The authors are not clear about where they believe Reg4/CFD mediated bactericidal activity is occurring. Their Fig.2 microbiota sequencing data and Fig.4 E. coli quantification data suggests that Reg4/CFD are acting on the luminal E. coli population, and the authors show detection of luminal CFD in Ext. data Fig.1C.

However, in their imaging data (Fig.2G, 6A) they focus on what appears to be tissue invasive *E. coli*. Where do the authors think this system is active? If the authors believe the activity is luminal then they must show imaging data supporting Reg4/complement binding to luminal *E. coli*.

Author Reply, 1) The activity of CFD is in the luminal. But, E.coli were analyzed under two different situations, normal and 2% DSS treated mice. In normal mice, we detected luminal E.coli population in feces; In 2% DSS treated mice, we detect E.coli in colon tissues such as Fig2G, Fig.6A.

2) For luminal E. coli, we have already shown imaging data supporting Reg4/complement binding to luminal E. coli (Figure 6a, b and Figure 7e)

Reviewer response: The authors need to be very clear at different points in the manuscript text and in the figure legends when they are analysing luminal and tissue associated E. coli, as it is possible that factors deposited on tissue invasive bacteria are not derived from the intestinal epithelium. I understand that the flow cytometry data in Fig. 6b and 7e indicate that these factors can be detected on luminal E. coli, but imaging data would still be useful to support this. I am unable to see any luminal E. coli in Fig. 6a, which the authors reference in their response.

Reply, High magnification images have been added into Fig. 6a.

7. The authors claim that increased E. coli in CFD and Reg4 KO mice is due to loss of bactericidal activity, supported by the fact that Reg4 and CFD can cooperatively enhance complement-mediated bacterial lysis in vitro. However, it is also quite possible that alterations in the mucosal environment in these mice modifies the growth of E. coli independent of bactericidal activity. The authors should try to show that bactericidal activity is actually altered in vivo, for example by quantifying live/dead E. coli cell ratios in vivo or by using an auxotrophic E. coli model that incapable of replicating in vivo.

Author Reply, 1) Indeed, it is also possible that alterations in the mucosal environment in these mice modify the growth of E. coli independent of bactericidal activity. To avoid this, we generated germ-free CFD KO and WT mice (Fig. 4). This should eliminate the alteration in these mice may modify the growth of E.coli.

2) It is hard to quantify live/dead E.coli cell ration in vivo. Although live/dead bacterial viability kits are available, it is only used in vitro for short time.

3) We will generate an auxotrophic E. coli model that is incapable of replicating in vivo in the future.

Reviewer response: The authors are correct that using GF mice removes the possibility that alterations in the microbiota might influence the mucosal environment; however, this does not correct for alterations in host-derived factors that might occur after loss of CFD or Reg4. Live/dead staining can be performed directly on bacteria

isolated from mouse faeces and be analysed by microscopy or flow cytometry. This would provide clear evidence of alterations in bactericidal activity *in vivo*.

Reply, We have analysed live/dead bacteria by flow cytometry. Now results have been shown in Figure 2 in this point by point reply. These results have also been added into Figure 4e and 7e in the manuscript respectively.

Figure 2. Both CFD KO and REG4 KO have reduced bactericidal ability. The equal amount of GFP-labeled *E.coli* (*E. coli* 0160, 1×10^7) were infused into different mice, including CTR1 (CFD^{fl/fl}pvillin-cre^wmice), CFDKO(CFD^{fl/fl}pvillin-cre^Tmice), CTR2 (WT) and REG4 KO mice. After 48 hrs, the feces from different mice were analysed using flow cytometry for live (GFP+PI-)/dead(GFP+PI+) bacteria. T-test, “***”, $P < 0.05$.

8. RE Fig.1 – The comparison in (A) should be performed by qRT-PCR. Is the CFDKO colon image (B) actually from colon? It looks more like small intestine.

Author Reply, 1) We performed qRT-PCR, and results was put in the Fig. 1A.
2) The CFDKO colon images indeed came from colon.

Reviewer response: The data in Fig. 1a looks fine; however, the CFDKO colon image is still quite confusing as it does look like small intestine in terms of crypt morphology and what look like villus structures. Can the authors please provide an alternative image that is less ambiguous?

Reply, Intestine and colon were confused. Now we have corrected them.

9. RE Ext. data Fig.1 – Presumably the GF mice are WT as well, so the comparison should be between ConvR and GF. In the Western blots in panel (F), what are the high molecular weight bands that appear in some of the samples?

*Author Reply, 1) Indeed, GF mice are from conventional wt mice.
2) The modified figures were put into supplementary Fig. 1.*

Reviewer response: As both GF and conventionally raised (ConvR) mice are genetically WT, please re-label graphs and figures with ConvR instead of WT. Regarding the high molecular weight bands in the Western blots in this figure, the intention was not for the authors to crop/remove this data but to identify and discuss it. Even if the authors do not determine the nature of these bands, their variable presence should be noted.

Reply, we have already used ConvR instead of WT; In addition, we can not determine the nature of high molecular weight bands.

10. RE Fig.2 – The microbiota data shown in (B-D) should be more clearly organized, e.g. charts should be labelled with genotype and treatment. Statistical comparison of microbiota data would also be useful, is it only the E. coli that is changing in the CFDKO mice? The antibody used in (E) is not detailed in the methods. The method for bacterial DNA extraction is not described.

*Author Reply, 1) We have already further organized the data in Fig. 2.
2) Statistical analyses have been done in Fig. 2.
3) FITC labeled anti- LPS antibody was described in supplementary Table S1.
4) The methods for bacteria DNA extraction have been added into method (gut microbiota analyses).*

Reviewer response: Bacterial DNA extraction methods have been expanded, but are still very unclear. For example, which ‘bacterial DNA extraction kit’ was used? Details for qPCR-based bacterial quantification (instrument, PCR conditions, PCR reagents etc.) are also absent.

Reply, The detailed method was added into methods (line 429-434)

11. RE Fig.3 – The differences between WT and CFDKO mice in the DSS experiments presented here are much less clear than the data shown in Fig.1, how reproducible is this phenotype? There should be statistical comparisons between the non-colonized and *E. coli* O160 colonized mice from both genotypes.

Author Reply, 1) As described in the manuscript, different concentrations of DSS were used. In figure 1, 2.5 % DSS was used; whereas 2% DSS was used in Figure 3 (Since colonization of *E.coli* potentially increases sensitivity to DSS-mediated colitis, 2% DSS was used in this experiment).

2) We have already performed statistical comparison between the non-colonized and *E. coli* O160 colonized mice between both genotypes (Fig. 3 in the manuscript)

Reviewer response: The rationale for using different concentrations of DSS in these experiments must be explicitly stated in the manuscript. I understand that the authors have presented statistical comparison between WT and CFDKO mice in Fig3a, d-g, but the group comparisons in Fig3b-c are not indicated. More importantly, the original request was for statistical comparison between non-colonized and *E. coli* O160-colonized mice from each genotype (e.g. WT vs WT/O160 and CFDKO vs CFDKO/O160). This is relevant as it is important to independently discriminate between the effects of genotype and *E. coli* colonization on DSS sensitivity.

Reply, The rationale for using different concentration of DSS have been stated in the text (line 161); Statistics have been added into Fig 3b-c.

12. RE Fig.4 – Are these colonization experiments still performed on animals undergoing DSS treatment as with all the previous experiments? If not, why not? There is no detail in methods regarding the generation of the GFP-*E. coli* strain, if the GFP label is plasmid based then in vivo plasmid stability quantification must be shown.

Author Reply, 1) No DSS-treatment. The goal of this experiment is to observe the act of CFD

and REG4 on the luminal bacteria, thus it was not necessary for DSS-treatment.

2) We added the method for the generation of the GFP-*E* into Fig. 4 legend.

Reviewer response: The rationale for not using DSS in this experiment must be explicitly stated in the text. Methods for GFP-*E. coli* generation should be described in detail in the methods section, not the figure legend. In vivo plasmid stability is quite variable, as bacteria tend to lose plasmids in the absence of selective pressure in vivo (e.g. during mouse colonization), do the authors have any data demonstrating plasmid stability during colonization?

Reply, We have explicitly stated the rationale for not using DSS in this experiment (line 199-200).

Method has been described for GFP-E. coli generation (line 450-453).

For plasmids stability, while GFP-E.coli were infused into germ-free mice, we did not found GFP-negative bacteria even after one week (Figure 3 in this point by point reply).

Figure 3. GFP-E. coli clones in the feces after colonization in germ-free mice. 1×10^7 GFP-E.coli were infused into germ free mice. GFP-E. coli clones in the feces of germ-free mice were checked on day 0, day 3 and day 7.

13. RE Ext. data Fig.4 – Blots shown in (C-E) require WT control images. Data from inflammasome-related KO tissues is difficult to interpret without littermate control samples.

Author Reply, We modified this figure and added control also (Complementary Fig. 4).

Reviewer response: The figure is improved, but there are still some issues. In the results text (line 168) Supplementary Fig.4a is referred to as qRT-PCR, but appears to be ELISA based on the legend and presentation in the figure, can the authors clarify this? Supplementary Fig.4c-e seem to show similar levels of IL-18 in vehicle treated WT and CFDKO tissues, but isn't this contradicted by Supplementary Fig.4a? Lastly, can the authors clarify if this data is from in vivo samples or tissues treated ex vivo, as in the original submission?

Reply, 1) we have already corrected qRT-PCR into ELISA. 2) We used confusing samples. Now, correct results have been added into supplementary Fig. 4. 3) Data in supplementary Fig. 4c-e are from tissues treated ex vivo. These have been added into figure legend.

14. RE Fig.5 – Control experiments with only the recombinant Reg4 constructs must be done to show that the protein has no intrinsic bactericidal activity. More detail regarding the Reg4 mutant must be provided, e.g. where is the protein mutated, does the mutation affect binding, etc.?

Author Reply, 1) Control experiments have already been done, relative data were added into Figure 5d.

2) Mutant hREGIV-P91S was described in methods (Generation of mouse Reg4 and human REG4 and mutant human REG4).

3) The reduced binding by mutated protein was shown (Fig. 5 in this point by point reply).

Reviewer response: Control experiments are convincing. Data shown in Fig.5 of the author reply does show a difference in WT and mutant Reg4 E. coli binding and this should be included in the manuscript. Could the authors comment on why Reg4 seems to only bind a small fraction of the E. coli cells in this data?

Reply, Only E. coli closed to gut wall are bound with Reg4.

15. RE Fig.6 – The use of antibiotic treatments in these experiments is not explained. Images from WT animals should be shown in (A). Binding experiment methods in (C-D) are not clearly detailed.

Author Reply, 1) The explain for use of antibiotic treatments have been added (line 379-380);

2) Images from WT animals have been shown in the Fig. 6 A in the manuscript;

3) Binding experiment methods have been further described in methods (Complement assays).

Reviewer response: The functional purpose of antibiotic use is obvious, but their use in this context needs to be explicitly stated in the manuscript text. Why is microbiota suppression required in this case? The inclusion of WT images in Fig.6 is useful, but the removal of high magnification images that were present in the original submission makes it impossible for the reader to evaluate the colocalisation of E. coli cells and the different proteins examined. High magnification images should be included for all panels.

Reply, 1) We have explicitly stated this in text (line 243-244); 2) High magnification images have been put into Figure 6a.

16. RE Fig.7 – Statistical comparison of microbiota data is needed. Are the Reg4 mice also more susceptible to DSS?

Author Reply, 1) We performed statistical comparison of microbiota data in the Fig. 7 in the manuscript;

2) Reg4 mice are more susceptible to DSS mediated colitis (Fig. 6 in this point by point reply). These data appear in another paper.

Microbiota comparison and Reg4 KO DSS susceptibility data is convincing.

REVIEWERS' COMMENTS:

Reviewer #3 (Remarks to the Author):

The authors have broadly addressed my concerns regarding specific aspects of the manuscript. The provision of total protein loading controls for Western blots is appropriate and makes this data more convincing. The provision of live/dead cytometric analysis also now gives a much clearer impression of CFD/Reg4-dependent bactericidal activity in vivo. Some small remaining points to correct are:

- 1) Please change all instances of 'WT' to 'ConvR' in Supplementary Figure 1.
- 2) The Reg4KO images of colon and small intestine seem to be the wrong way around in Supplementary Figure 6.